# Simultaneous Momentum and Position Measurement and the Instrumental Weyl-Heisenberg Group

**DOI:** 10.3390/e25081221

**Published:** 2023-08-16

**Authors:** Christopher S. Jackson, Carlton M. Caves

**Affiliations:** 1Independent Researcher, Gold Beach, OR 97444, USA; omgphysics@gmail.com; 2Center for Quantum Information and Control, University of New Mexico, Albuquerque, NM 87131, USA

**Keywords:** Weyl-Heisenberg group, measuring instrument, Kraus operator, Cartan decomposition, Harish-Chandra decomposition, right-invariant derivative, Maurer-Cartan form, Wiener path integral, diffusion equation, stochastic differential equation

## Abstract

The canonical commutation relation, [Q,P]=iℏ, stands at the foundation of quantum theory and the original Hilbert space. The interpretation of *P* and *Q* as observables has always relied on the analogies that exist between the unitary transformations of Hilbert space and the canonical (also known as contact) transformations of classical phase space. Now that the theory of quantum measurement is essentially complete (this took a while), it is possible to revisit the canonical commutation relation in a way that sets the foundation of quantum theory not on unitary transformations but on positive transformations. This paper shows how the concept of simultaneous measurement leads to a fundamental differential geometric problem whose solution shows us the following. The simultaneous *P* and *Q* measurement (SPQM) defines a universal measuring instrument, which takes the shape of a seven-dimensional manifold, a universal covering group we call the instrumental Weyl-Heisenberg (IWH) group. The group IWH connects the identity to classical phase space in unexpected ways that are significant enough that the positive-operator-valued measure (POVM) offers a complete alternative to energy quantization. Five of the dimensions define processes that can be easily recognized and understood. The other two dimensions, the normalization and phase in the center of the IWH group, are less familiar. The normalization, in particular, requires special handling in order to describe and understand the SPQM instrument.

## 1. Introduction

After World War II, theoretical quantum physics became dominated by the design of quantum field theory. There were three branches of physics that stemmed from this: high energy, condensed matter, and atomic-molecular-optical (AMO) physics. Although incredibly developed as predictive methods, quantum field theory in all three of these branches has left some very basic ideas of quantum observation underdeveloped. That this is indeed still the case is evident in how the coherent-state resolutions of the identity or “overcomplete bases” [1,2,3,4] are usually finessed:(1)$$1_{Z}^{\left(\mathrm{amp}\right)}=Z\int_{C}\frac{d^{2}\alpha }{\pi }|\sqrt{Z}\alpha \rangle \;\langle \sqrt{Z}\alpha |$$
for bosonic amplitudes (where *Z* is an arbitrary complex scalar) [1,5,6,7,8] and
(2)$$1_{j}^{\left(\mathrm{spin}\right)}=\left(2j+1\right)\int_{S^{2}}\frac{d\mu (\hat{n})}{4\pi }|j,\hat{n}\rangle \;\langle j,\hat{n}|$$
for fermionic spins (where 2j is an integer) [2,3,4,9]. These overcomplete bases are key both to establishing functional integration [1,10,11,12,13,14,15,16,17,18,19] and to understanding the observation of energy quanta [20]. Yet these overcomplete bases do not fit the most basic idea of a Hermitian eigenbasis and as such are often not considered a serious form of observation. By itself, this inattention to how one could observe in these overcomplete bases leaves a missing piece at the foundation of quantum mechanics and quantum field theory.

Meanwhile, an understanding of the overcomplete bases as a bona fide means of observation has been slowly coming to light. First, the idea of observables matured into the general mathematical theories of instruments and operations [21,22,23,24,25,26,27,28,29,30,31,32]. In this more contemporary language, the overcomplete bases of Equations (Equation 1) and (Equation 2) are called *positive-operator-valued measures* (POVMs) and are understood to be informationally complete, meaning the distribution observed with any one of these POVMs is enough to reconstruct the quantum state. With this theoretical technology, it has further been discovered that the overcomplete bases correspond to various forms of continual (or continuous) simultaneous observation. The standard coherent-state POVM given in Equation (Equation 1) was discovered to be the effect of simultaneously observing both quadratures of a leaky cavity, a form of observation we will call Goetsch-Graham-Wiseman (GGW) heterodyne detection [33,34,35,36,37]. Then, the spin-coherent POVM of Equation (Equation 2) was discovered to be the effect of simultaneously observing the three orthogonal spin components, a form of observation we call the *spin-isotropic measurement* [38,39,40].

Before proceeding, we caution that this paper uses a mathematical apparatus not familiar to most physicists and quantum scientists. This apparatus is introduced here naturally as it becomes both desirable and necessary. Readers who are made uncomfortable by this apparatus are urged to consult the companion paper [40], which attempts to persuade the reader that the unfamiliar mathematical concepts and techniques are essential tools—a new way of thinking and doing—and then introduces these tools as gently as possible.

GGW heterodyne detection and the spin-isotropic measurement work in a very similar way, but they are different in one very important respect. While GGW heterodyne detection assumes energy-conserving system-meter interactions, the spin-isotropic measurement assumes Hermitian meter displacements, $-iH^{\left(\mathrm{iso}\right)}dt/\hbar =-\sqrt{\kappa \;dt}\;J_{k}⊗2\sigma \partial_{q}$, where $J_{k}$ is an orthogonal spin component of the system, *q* is the meter register, σ is the width of the meter pointer, and κ is the measurement rate. In both cases, the measuring instrument consists of Kraus operators defined by a time-ordered exponential over the duration *T* of the measurement. For GGW heterodyne detection, the Kraus operators are [37]
(3)$$L^{\left(\mathrm{GGW}\right)}\left[dw_{\left[0,T\right)}\right]=T\exp\;\left(\int_{0}^{T-dt}-2a^{†}a\;\kappa \;dt+2a\sqrt{\kappa }\;dw_{t}^{*}\right)\;,$$
where $a=\left(Q+iP\right)/\sqrt{2\hbar }$ is the usual complex-amplitude operator and $dw_{t}=\left(dW_{t}^{q}+idW_{t}^{p}\right)/\sqrt{2}$ is the registered complex Wiener path. For spin-isotropic measurement, the Kraus operators are [39,40]
(4)$$L^{\left(\mathrm{iso}\right)}\left[d{\vec{W}}_{\left[0,T\right)}\right]=T\exp\;\left(\int_{0}^{T-dt}-{\vec{J}}^{\;2}\;\kappa \;dt+\vec{J}\cdot \;\sqrt{\kappa }\;d{\vec{W}}_{t}\right)\;,$$
where $\vec{J}=\left(J_{x},J_{y},J_{z}\right)$ is the triple of orthogonal spin-component observables and $d{\vec{W}}_{t}=\left(dW_{t}^{x},dW_{t}^{y},dW_{t}^{z}\right)$ is the registered three-vector of Wiener paths. The most striking feature about the instruments defined by Equations (Equation 3) and (Equation 4) is that they can be integrated *universally*, that is, independently of matrix representation. The difference between the two cases can now be summarized as the following: integrating Equation (Equation 4) defines a seven-dimensional manifold that requires the theory of symmetric spaces, whereas integrating Equation (Equation 3) defines a three-dimensional manifold that is much more straightforward.

This paper is an analysis of the quadrature analog of the spin-isotropic measurement, a form of observation we call the *simultaneous P and Q measurement* (SPQM). The name SPQM is our homage to Alberto Barchielli, who appears to be the first to have considered and analyzed this problem [41] (*Senatus PopulusQue Romanus* (SPQR), translated as “The Senate and People of Rome”, is an enduring symbol of ancient Rome [42]. SPQM has been similarly associated with the city of Milan, so we offer it as a tribute to Barchielli and his lifetime of work on continuous measurement theory at the University of Milan). SPQM generates a measuring instrument with Kraus operators [40]
(5)(L(SPQM)dw[0,T)=Texp∫0T−dt−2Hoκdt+PκdWtp+QκdWtq,)
where $2H_{o}\equiv P^{2}+Q^{2}$ and *P* and *Q* are (dimensionless) canonical momentum and position (or the conserved quadrature components of a harmonic oscillator). This time-ordered exponential defines another fundamental seven-dimensional manifold, the universal covering group, which for SPQM we call the *instrumental Weyl-Heisenberg group*, G=IWH. The universal covering group is defined by a map γ with the universal property that for any Hilbert space H carrying the paths of Kraus operators $L^{\left(\mathrm{SPQM}\right)}:C^{T/dt}⟶\mathrm{GL}\left(H\right)$, there exists a unique representation *R* such that
(6)$$L^{\left(\mathrm{SPQM}\right)}=R\circ \gamma \;\mathrm{where}\;C^{T/dt}\overset{\;\gamma \;}{\to }\mathrm{IWH}\overset{\;R\;}{\to }\mathrm{GL}\left(H\right)\;.$$
This universal way of considering G=IWH essentially amounts to suspending the choice of H and, therefore, *ℏ*, but it is very important to appreciate that the measuring instrument is, in fact, fundamentally independent of the Hilbert space and, therefore, *ℏ*. The same is true for the spin-isotropic measurement, except that different irreducible representations do not amount to choices of *ℏ*, but rather to choices of the total angular momentum number *j*.

The universal covering group IWH can be navigated in much the same way as can be done for semisimple Lie groups, with the use of right-invariant vector fields and decompositions similar to those of Cartan and Harish-Chandra. In particular, the sample paths defined by $x\left(t\right)=\gamma \left[dw_{\left[0,t\right)}\right]$ diffuse according to a Fokker-Planck-Kolmogorov equation,
(7)(1κ∂∂tDt(x)=2Ho←+12P←P←+12Q←Q←[Dt](x),)
where
(8)(DT(x)≡∫Dμdw[0,T)δx,γdw[0,T))
is the *Kraus-operator distribution function* of the SPQM instrument with respect to the Haar measure [18,43,44,45] of the IWH group and $\underset{\leftarrow }{H_{o}}$, $\underset{\leftarrow }{Q}$, and $\underset{\leftarrow }{P}$ are right-invariant derivatives tangent to the IWH group. We regard “Kraus-operator distribution function”, “Kraus-operator distribution”, and “Kraus-operator density” as interchangeable, despite subtle differences some might attribute to these usages. We abbreviate Kraus-operator distribution function as KOD to invite the reader to use any of these terms. The KOD can be considered the *universal unraveling* of the total (or unconditional) operation (a completely positive, trace-preserving superoperator),
(9)(ZT(SPQM)≡∫Dμdw[0,T)O·L(SPQM)dw[0,T)=∫IWHd7μ(x)DT(x)O·R(x),)
where $D\mu \;\left[dw_{\left[0,T\right)}\right]$ is the Wiener path measure, $d^{7}\;\mu \left(x\right)$ is the Haar measure, and $O\;\cdot \;\left(L\right)\equiv L\;⊙L^{†}$. The technology of right-invariant differentiation [46,47,48] will not be familiar to most quantum physicists and information scientists. Introducing SPQM, the IWH group, the concept of right-invariant motion, and the KOD is the subject of Section 2.

Section 3 translates the coordinate-independent formulation from Section 2 to forms that physicists are more likely to recognize. Indeed, what the aforementioned decompositions do is to coordinate the points of the IWH group [2,3,49,50,51,52]. The decomposition of the IWH group, similar to Harish-Chandra [52,53,54] (also known as “Gauss” [2,51,55]) decomposition, is given by
(10)$$x=e^{a^{†}\nu }e^{-H_{o}r+\Omega z}e^{a\mu^{*}}\;,$$
with the purity coordinate r∈R, which we will call the ruler; central coordinate z=−s+iψ∈C; phase-space coordinates ν,μ∈C; and $\Omega =\hbar 1_{H}$. The decomposition of the IWH group, similar to the Cartan decomposition [50,51,52,54], is given by
(11)$$x=D_{\beta }e^{i\Omega \phi }e^{-H_{o}r-\Omega \ell }D_{\alpha }^{\;-\;1}\;,$$
with different central coordinates ϕ,ℓ∈R, the same ruler *r*, and phase-space coordinates β,α∈C appearing in the conventional displacement operator $D_{\alpha }\equiv e^{a^{†}\alpha -a\alpha^{*}}$. Introducing these decompositions and using them to transform Equation (Equation 5) into standard Itô-form stochastic differential equations (SDEs) [56,57,58,59,60] and to transform Equation (Equation 7) into a coordinate Fokker-Planck-Kolmogorov (FPK) diffusion equation [59,60] is the subject of Section 3, which also solves those SDEs and (mostly) solves the FPK diffusion equation.

As a function of the registers of SPQM, we find that the ruler satisfies
(12)$$r_{T}=2\kappa T.$$
For the remaining Harish-Chandra coordinates, we find that the phase points follow Ornstein-Uhlenbeck [59,60,61] and GGW processes,
(13)$$\nu \left[dw_{\left[0,T\right)}\right]=\int_{0}^{T_{-}}\;\;\sqrt{\kappa }\;dw_{t}\;e^{-2\kappa (T-t)}\;\mathrm{and}\;\mu \left[dw_{\left[0,T\right)}\right]=\int_{0}^{T_{-}}\;\;\sqrt{\kappa }\;dw_{t}\;e^{-2\kappa t}\;,$$
where $T_{-}\equiv T-dt$, and the center follows a quadratic functional process,
(14)$$z\left[dw_{\left[0,T\right)}\right]=\frac{1}{2}\int_{0}^{T_{-}}\;\;\int_{0}^{T_{-}}\;\;\kappa \;dw_{t}^{*}\;dw_{s}\;e^{-2\kappa |t-s|}\left(1+\mathrm{sgn}(t-s)\right)\;,$$
where the sign function is used: sgn(u)=u/|u| for u≠0 and sgn(0)=0. The Cartan phase-space coordinates follow linear functionals,
(15)$$\beta \left[dw_{\left[0,T\right)}\right]=\int_{0}^{T_{-}}\sqrt{\kappa }\;dw_{t}\;\frac{\cosh2\kappa t}{\sinh2\kappa T}\;\mathrm{and}\;\alpha \left[dw_{\left[0,T\right)}\right]=\int_{0}^{T_{-}}\sqrt{\kappa }\;dw_{t}\;\frac{\cosh2\kappa (T-t)}{\sinh2\kappa T}\;.$$
The Cartan central coordinates *ℓ* and ϕ follow from the coordinate transformation between Harish-Chandra and Cartan coordinates, which is given in Equations (A34), (A35), (Equation 370) and (A39), but we do not write those solutions explicitly here.

As for the KOD $D_{t}\left(x\right)$, we will not be able to solve analytically for the distribution over all seven dimensions. Summing over the center
(16)$$Z\equiv \left\{e^{1z}\;:\;z\in C\right\}⊲\mathrm{IWH}\;,$$
however, gives a *reduced SPQM unraveling* of the total operation,
(17)$$\begin{array}{c} Z_{T}^{\left(\mathrm{SPQM}\right)}=\int_{\mathrm{IWH}/Z}d^{5}\;\mu \left(Zx\right)\;C_{T}\left(Zx\right)\;O\;\cdot \;\;\left(D_{\beta }\;e^{-H_{o}r}D_{\alpha }^{†}\right)\;, \end{array}$$
where the integral is over the quotient group IWH/Z, which we call the *reduced instrumental Weyl-Heisenberg* (RIWH) group. The reduced Kraus-operator distribution (RKOD),
(18)(CT(Zx)≡∫ZdϕdℓDT(x)e−2ℓ=∫Dμdw[0,T)e−2ℓ[dw[0,T)]δZx,Zγdw[0,T),)
is a marginal over the center that includes the Cartan center factor $e^{-2\ell }$. We call $C_{T}\left(Zx\right)$ the *Cartan-section reduced distribution function*, and we are able to solve for it from its FPK diffusion equation. The solution is a Gaussian with ill-defined normalization,
(19)$$d^{5}\;\mu \left(Zx\right)\;C_{T}\left(Zx\right)=2\sinh r\;dr\;\delta \left(r-2\kappa T\right)\frac{d^{2}\beta }{\pi }\frac{1}{\Sigma_{T}}e^{{-|\beta -\alpha |}^{2}/\Sigma_{T}}\frac{d^{2}\alpha }{\pi }\;,$$
where the mean-square distance between the two phase points is given by
(20)$$\Sigma_{T}=\kappa T-\tanh\kappa T\;.$$
The normalization factor 2sinh2κT is particularly interesting, as the POVM completeness relation for the SPQM instrument boils down to (assuming ℏ=1)
(21)$$2\sinh2\kappa T\int \;\frac{d^{2}\alpha }{\pi }\;D_{\alpha }\;e^{-H_{o}4\kappa T}D_{\alpha }^{†}=1_{H}\;,$$
and this can be recognized as equivalent to the result of energy quantization,
(22)$$\mathrm{tr}e^{-H_{o}4\kappa T}=\sum_{n=0}^{\infty }e^{-(n+\frac{1}{2})4\kappa T}=\frac{1}{2\sinh2\kappa T}\;.$$

This demonstrates that the KOD can be considered an alternative to energy quantization. Indeed, the operator $H_{o}$ does not appear in SPQM as an energy observable, but rather as a term required by the positivity of sampling measurement records. The late-time limit of the completeness relation is of particular interest: when T≫1/κ, $D_{\alpha }\;e^{-H_{o}4\kappa T}D_{\alpha }^{†}=e^{-2\kappa T}\left|\alpha \rangle \;\langle \alpha \right|$, showing that the SPQM POVM elements approach Glauber coherent states; the completeness relation shows that it does so uniformly,
(23)$$\begin{array}{c} 1_{H}=\int \frac{d^{2}\alpha }{\pi }|\alpha \rangle \;\langle \alpha |\;, \end{array}$$
thus giving the coherent-state resolution of the identity.

Everything in our analysis—the path integrals, the KODs, the FPK diffusion equations, the SDEs—follows from the path integral for the overall quantum operation $Z_{T}^{\left(\mathrm{SPQM}\right)}$, given in Equation (Equation 9), which integrates over the sample paths as they appear in the time-ordered Kraus operator $L^{\left(\mathrm{SPQM}\right)}\left[dw_{\left[0,T\right)}\right]$. As the analysis develops in Section 2 and Section 3, however, there emerges a disconnect between the reduced distribution $C_{T}\left(Zx\right)$, which expresses the completeness relation, and the SDEs and stochastic integrals for the phase-point coordinates: $C_{t}\left(Zx\right)$ has ill-defined normalization, and it is not the weight function whose moments are those of the Cartan phase-point variables, β and α, as they are expressed in the stochastic-integral solutions of Equation (Equation 15). The disconnect is all about the real center term, $e^{-\Omega \ell }$, which scales the Kraus operators when they are written in Cartan coordinates.

The three faces of the KOD stochastic trinity—path integrals, diffusion equations, and SDEs—having been sundered in Section 3, are re-united in Section 4. The vehicle for the reunion is the *Harish-Chandra-section reduced distribution function*,
(24)(BT(Zx)≡∫ZdψdsDT(x)e−2s=∫Dμdw[0,T)e−2s[dw[0,T)]δZx,Zγdw[0,T),)
where −s is the real part of the Harish-Chandra center coordinate *z*. The distribution $B_{T}\left(Zx\right)$ can be determined in two equivalent ways: first, from the FPK diffusion equation for $B_{T}\left(Zx\right)$ and, second, from applying the stochastic integrals for the Harish-Chandra phase-plane variables, ν and μ, to the above path-integral expression for $B_{T}\left(Zx\right)$. The stochastic trinity thus restored, one returns to completeness through the relation
(25)$$C_{T}\left(Zx\right)=e^{f(Zx)}B_{T}\left(Zx\right)\;,$$
where f(Zx) is a quadratic function of the phase-plane variables, given in Equations (Equation 362) and (Equation 370), which comes directly out of the coordinate transformation between Cartan and Harish-Chandra coordinates; $e^{2f(Zx)}$ can be regarded as a positive gauge transformation [19].

Section 5 concludes with musings on the stochastic trinity and the Lie-group manifolds that house instrument evolution.

## 2. The IWH Group and Coordinate-Free Right-Invariant Motion

This section defines the *simultaneous P and Q measurement* (SPQM) process and presents the instrumental Weyl-Heisenberg group, G=IWH, as the universal covering group of SPQM.

Section 2.1 introduces Kraus operators and the concept of observables generating positive transformations instead of unitary transformations. Section 2.2 introduces the SPQM process and the group IWH. Section 2.3 explains how the SPQM instrument is universal and defines a *Kraus-operator distribution (or density)* (KOD) over the IWH group. Section 2.4 explains how the KOD diffuses over time with the introduction of *right-invariant derivatives*, a differential-geometric technology that will be unfamiliar to most physicists and quantum scientists. Section 2.5 explains how the sample paths of the Kraus-operator diffusion are described with the introduction of the *right-invariant one-forms*, which are duals of right-invariant derivatives.

The diffusion equation and stochastic differential equations in Section 2.4 and Section 2.5 are solved in Section 3.

### 2.1. Observables and Infinitesimal Positive Transformation

While observables are often considered to be infinitesimal generators of unitary transformations, they can also generate *positive transformations*. Let *X* be a Hermitian observable, κ be a real number with units of inverse time, and dW be a standard Wiener increment [10,11,12,15,16,19,59,60], which has the measure
(26)(dμ(dW)=d(dW)2πdte−dW2/2dt.)
Unitary transformations can be infinitesimally generated either deterministically or stochastically, such as
(27)$$e^{-iX\kappa \;dt}\;\mathrm{or}\;\sqrt{d\mu (dW)}e^{-iX\sqrt{\kappa }\;dW}\;.$$
Positive transformations, on the other hand, are fundamentally stochastic. In addition, the infinitesimal generators of positive transformations are not of the canonical form, with a single parameter conjugate to the infinitesimal generator. The positive transformations we will be interested in do not involve jump operators, as in photodetection [37,62], but rather are differential, with infinitesimal generators of the form [40,63,64,65]
(28)(dμ(dW)LX(dW)≡dμ(dW)e−X2κdt+XκdW.)
As a set, these positive transformations define a *measuring instrument*, complete over the Hilbert space, H, according to the relation,
(29)$$\int_{R}d\mu \left(dW\right)L_{X}{\left(dW\right)}^{†}L_{X}\left(dW\right)=1_{H}\;;$$
the operators $L_{X}\left(dW\right)$ are known as *Kraus operators* [29,31], and the set of elements, $\left\{d\mu \left(dW\right)L_{X}{\left(dW\right)}^{†}L_{X}\left(dW\right)\right\}$, is known as a *positive-operator-valued measure* (POVM). We will call the positive transformations of Equation (Equation 28) *differential Kraus operators* and their set a *differential instrument*, both in anticipation of the (multi-dimensional) differential geometry coming up and to contrast these Kraus operators with jump operators.

The form of these infinitesimally generated positive transformations comes from the requirement that the total operation be completely positive and trace preserving. In particular, we can define the superoperator A⊙B by
(30)A⊙B(ρ)≡AρB
and the (selective, Kraus-rank-one) *operations* by
(31)$$O\;\cdot \;\left(L\right)\equiv L\;⊙\;L^{†}\;.$$
Then, the aforementioned Kraus operators define a *total operation*,
(32)$$\int d\mu \left(dW\right)\;O\;\cdot \;\;\left(e^{-X^{2}\kappa dt+X\sqrt{\kappa }\;dW}\right)=e^{-\frac{1}{2}\kappa \;dt\;{\mathrm{ad}}_{X}^{2}}\;,$$
where the *adjoint* superoperator is defined by
(33)$${\mathrm{ad}}_{X}\equiv X⊙1-1⊙X\;.$$
In this context, the infinitesimal generator or *Lindbladian*,
(34)$$-\frac{1}{2}{\mathrm{ad}}_{X}^{2}=X⊙X-\frac{1}{2}\left(X^{2}⊙1+1⊙X^{2}\right)\;,$$
defines *X* as a *Lindblad operator* [28].

Kraus operators can be interpreted as an *indirect measurement* [29,31,40,66,67] where, in the differential case, a meter with an initial meter wavefunction
(35)$$\sqrt{dq}\;\langle q|0\rangle =\sqrt{\frac{dq}{\sqrt{2\pi \sigma^{2}}}e^{-q^{2}/2\sigma^{2}}}\;,$$
is displaced by the system according to the interaction
(36)$$-\frac{i}{\hbar }Hdt=\sqrt{\kappa \;dt}\;X⊗\;\left(-2\sigma \frac{\partial }{\partial q}\right)\;,$$
which, in turn, registers some “position” *q*, fixing
(37)$$dW=\sqrt{dt}\;\frac{q}{\sigma }\;,$$
so that
(38)$$\sqrt{d\mu (dW)}\;L_{X}\left(dW\right)=\sqrt{dq}\;\langle q|e^{-iH\;dt/\hbar }|0\rangle$$
(for further details, see [40]).

An irresistible sidenote, developed more generally and in more detail in [40], is that the stochastic unitary transformations of Equation (Equation 27) follow from the same meter interaction, given in Equation (Equation 36), but with registration of the meter momentum *p* instead of its position *q*. As such, the stochastic unitary transformations have a total operation identical to Equation (Equation 32),
(39)$$\int d\mu \left(dW\right)\;O\;\cdot \;\;\left(e^{-iX\sqrt{\kappa }\;dW}\right)=e^{-\frac{1}{2}\kappa \;dt\;{\mathrm{ad}}_{X}^{2}}\;,$$

This alternative unraveling of the total operation corresponds to a symmetry of the general Lindbladian,
(40)$$L\left(A\right)=A⊙A^{†}-\frac{1}{2}\left(A^{†}A⊙1+1⊙A^{†}A\right)\;,$$
which is
(41)L(−iX)=L(X),
so that −iX is an alternative Lindblad operator of the total operation.

### 2.2. SPQM and the IWH Group

The subject of this paper is the continual (or continuous) simultaneous observation of the canonical observables *P* and *Q* defined by the canonical commutation relations,
(42)([Q,P]=iΩand[Ω,Q]=[Ω,P]=0.)
As a Lie algebra of infinitesimal generators, these observables are usually considered to generate unitary displacement operators,
(43)$$D_{\alpha }=e^{iQ\alpha_{2}-iP\alpha_{1}}\;,\;\mathrm{where}\;\alpha =\frac{\alpha_{1}+i\alpha_{2}}{\sqrt{2}}\;,$$
which together define the three-dimensional *unitary Weyl-Heisenberg group*,
(44)(K=UWH≡DαeiΩϕ:α∈C,ϕ∈R.)
If the generators operate irreducibly on the Hilbert space H, then, by Shur’s lemma, $\Omega =\hbar 1_{H}$ for some ℏ∈R (because $g\Omega g^{\;-\;1}=\Omega$ for all g∈UWH.) Assuming that *ℏ* is finite—that is, ℏ≠0—all such representations are essentially equivalent because the observables can always be rescaled so as to make the choice ℏ=1. Therefore, it is usually assumed that
(45)(Ω=1H.)
We shall now assume that ℏ=1 but we will continue nonetheless to use Ω for the infinitesimal generator to emphasize that its existence is not defined by the Hilbert space but instead by the canonical commutation relations. In particular, what is defined by the Hilbert space is the relation $\Omega^{2}=\Omega$, which is associative algebraic and not Lie algebraic.

For quantization, the unitary Weyl-Heisenberg group is supplemented with the unitary group generated by
(46)$$H_{o}\equiv \frac{P^{2}+Q^{2}}{2}\;,$$
defining the four-dimensional *dynamical Weyl-Heisenberg group*,
(47)$$\mathrm{DWH}\equiv \left\{e^{-iH_{o}s}D_{\alpha }\;e^{i\Omega \phi }\;:\;s\in R,\;\alpha \in C,\;\phi \in R\right\}\;.$$
Here, the use of “dynamical” refers to the analogies between $H_{o}$ and the classical Hamiltonian of a simple harmonic oscillator, upon which quantum mechanics was originally founded.

The observables *P* and *Q* can be measured simultaneously in the sense that the positive transformations they generate commute infinitesimally (to order dt),
(48)L(dw)≡LQdWqLPdWp(49)=e−Q2κdt+QκdWqe−P2κdt+PκdWp(50)=e−(Q2+P2)κdt+QκdWq+PκdWp+12[Q,P]κdWqdWp(51)=e−(Q2+P2)κdt+QκdWq+PκdWp(52)=LPdWpLQdWq,
so long as the Wiener outcome increments, $dW^{q}$ and $dW^{p}$, are independent, their joint Wiener measure being
(53)$$d\mu \left(dw\right)\equiv d\mu \left(dW^{q}\right)d\mu \left(dW^{p}\right)=\frac{d^{2}\left(dw\right)}{\pi dt}e^{-dw^{*}dw/dt}\;.$$
Here we switch to using a complex Wiener increment
(54)$$dw\equiv \frac{dW^{q}+idW^{p}}{\sqrt{2}}\;.$$

Continually repeating this simultaneous measurement for a finite amount of time, *T*, defines the overall Kraus operators,
(55)$$L\;\left[dw_{\left[0,T\right)}\right]=T\;\;\prod_{k=0}^{T/dt-1}L\left(dw_{kdt}\right)\;,$$
where T∏ denotes a time-ordered product. It is important to appreciate that, while the Kraus operators commute infinitesimally, they do not commute over finite amounts of time. Finally, these Kraus operators are accompanied by the Wiener path measure,
(56)$$D\mu \;\left[dw_{\left[0,T\right)}\right]=\prod_{k=0}^{T/dt-1}\;d\mu \left(dw_{kdt}\right)=\left(\prod_{k=0}^{T/dt-1}\;d^{\;2}\;\left(dw_{kdt}\right)\right){\left(\frac{1}{\pi dt}\right)}^{T/dt}\exp\;\left(-\int_{0}^{T_{-}}\frac{|dw_{t}{|}^{2}}{dt}\;\right)\;,$$
which is written here in terms of the complex Wiener increments. In summary, we have defined a time-dependent instrument with Kraus operators
(57)(Dμdw[0,T)Ldw[0,T)=Dμdw[0,T)Texp∫0T−Ho2κdt+QκdWtq+PκdWtp,)
where Texp∫ is the time-ordered exponential. This is the *simultaneous P and Q measurement* (SPQM.) It is worth repeating that $H_{o}$ appears here due to the form of the differential positive transformations of Equation (Equation 28) and in this context is not a Hamiltonian because it is not generating unitary transformations.

While SPQM has been considered as far back as [41], it has not been fully solved before. There are two other ways of measuring *P* and *Q* simultaneously that are important to distinguish from SPQM: the Arthurs-Kelly measurement [67,68] and the Goetsch-Graham-Wiseman (GGW) model of heterodyne detection [33,34,35,36,37]. The Arthurs-Kelly measurement has the same system-meter interaction as the SPQM process but is different in that Arthurs and Kelly imagine continually interacting the same two meters with the system until the measurement is terminated, whereas in the SPQM process, the system interacts with many pairs of meters successively, registering the complex Wiener path $dw_{\left[0,T\right)}$. The GGW model of heterodyne detection has the same many-meter model as the SPQM process, but each system-meter interaction is energy-conserving (the so-called leaky cavity), whereas the system-meter interaction of the SPQM process is the meter displacement given in Equation (Equation 36).

The total operation of the SPQM process is a Wiener-like path integral,
(58)$$Z_{T}\equiv \int D\mu \;\left[dw_{\left[0,T\right)}\right]O\;\cdot \;\left(L\;\left[dw_{\left[0,T\right)}\right]\;\right)\;,$$
which is absolutely trivial to solve,
(59)$$Z_{T}={\left(\int d\mu \left(dw\right)\;O\;\cdot \;\left(e^{-H_{o}2\kappa dt+Q\sqrt{\kappa }dW_{t}^{q}+P\sqrt{\kappa }dW_{t}^{p}}\right)\right)}^{\circ T/dt}=e^{-\frac{1}{2}\kappa T\left({\mathrm{ad}}_{Q}^{2}+{\mathrm{ad}}_{P}^{2}\right)}\;.$$
The interest of this article, however, is entirely in the manifold diffusion process defined by SPQM, where Equation (Equation 57) is understood to define sample paths in a finite-dimensional manifold. The infinitesimal generators of these sample paths are *Q*, *P*, and $H_{o}$. By simply considering their Lie brackets to the first order,
(60)[Q,P]=iΩ,(61)o,Q]=−iP,(62)o,P]=iQ,
and second order,
(63)[Ho,Q],Q=−Ω,(64)[Ho,P],P=−Ω,(65)Ho,[Ho,Q]=Q,(66)Ho,[Ho,P]=P,
we can see that SPQM defines a seven-dimensional manifold, a representation of what we will call the *instrumental Weyl-Heisenberg group*,
(67)(G=IWH≡e−HoreQqePpDαeiΩϕe−Ωℓ:r∈R,q,p∈R,α∈C,ϕ,ℓ∈R.)

There is a fourth and final Weyl-Heisenberg group worth defining, the six-dimensional *complex Weyl-Heisenberg group*,
(68)$$C\mathrm{WH}\equiv \left\{e^{Qq}e^{Pp}e^{-\Omega \ell }D_{\alpha }\;e^{i\Omega \phi }\;:\;q,p\in R,\alpha \in C,\phi ,\ell \in R\right\},$$
which is a maximal normal subgroup of the IWH group called the derived subgroup. In Lie-group terminology, the IWH group is said to be solvable while CWH is said to be nilpotent or unipotent [51,52]. In particular, the derived series of *G* is
(69)G=IWH⊳CWH⊳Z⊳1,
where the *center* of *G* is defined,
(70)(Z≡eiΩϕe−Ωℓ:ϕ,ℓ∈R.)

Before proceeding, it is useful to introduce the complex-amplitude (also known as annihilation) operator
(71)$$\begin{array}{c} a\equiv \frac{1}{\sqrt{2}}\left(Q+iP\right)\;, \end{array}$$
which has the Lie bracket
(72)$$\begin{array}{c} {}[a,a^{†}]=\Omega \;. \end{array}$$
We have
(73)$$\begin{array}{c} H_{o}=\frac{1}{2}\left(aa^{†}+a^{†}a\right)=a^{†}a+\frac{1}{2}\Omega \;. \end{array}$$
Note also that the displacement operator,
(74)$$\begin{array}{c} D_{\alpha }=e^{iQ\alpha_{2}-iP\alpha_{1}}=e^{a^{†}\alpha -a\alpha^{*}}\;, \end{array}$$
has the ordered forms,
(75)Dα=e−12iΩα1α2eiQα2e−iPα1=e12iΩα1α2e−iPα1eiQα2(76)=e−12|α|2Ωea†αe−aα*=e12|α|2Ωe−aα*ea†α,
which will prove useful in relating coordinate systems and in evaluating right-invariant derivatives. The first form in Equation (76) is usually called normal ordering, and the second form is called antinormal ordering.

### 2.3. Haar Measure, Dirac Delta, and Kraus-Operator Distribution Function

Many readers will interpret the groups defined in the previous section as matrix groups, where the observables are quantized in the usual way. The exponentials can be understood more abstractly, however, as generating path-connections, and this way of thinking gives rise to what is called the *universal covering group* [40,52,69,70,71,72]. We will now start to consider more seriously the points of the IWH group in this universal fashion and think of the instrument of SPQM as a representation of G=IWH. The map $L:C^{T/dt}⟶\mathrm{GL}\left(H\right)$ from the set of paths, $C^{T/dt}$, to the operator space, GL(H), can be factored into two maps,
(77)(L=R∘γ,)
where $\gamma :C^{T/dt}⟶G$ maps Wiener paths to the universal covering group and R:G⟶GL(H) is the representation, mapping the universal cover to the space of linear operators on H [52]. We have denoted a sample path by $dw_{\left[0,T\right)}$, and we now start denoting elements of the instrumental group by *x*. To drive the notation home, we note that the instrumental group element and Kraus operator associated with a sample path are denoted by
(78)$$\begin{array}{c} x_{T}=\gamma \left[dw_{\left[0,T\right)}\right]\;\;\mathrm{and}\;\;L\left[dw_{\left[0,T\right)}\right]=R\left(x_{T}\right)=R\left(\gamma [dw_{\left[0,T\right)}]\right)\;. \end{array}$$
The distinction between *L* and γ emphasizes that the time-ordered exponential of Equation (Equation 57) actually defines a diffusion problem on the instrumental group that is independent of the spectral information inherent in the definition of a linear operator. In particular, this means the entire analysis of this article is independent of whether or how $H_{o}$ is quantized (remember that $H_{o}$ is quantized in the usual way the moment H is assumed to be irreducible and to have a ground state).

As with every finite-dimensional Lie group, G=IWH has a right-invariant *Haar measure* [18,43,44,45],
(79)(dμ(xg)=dμ(x).)
As is *not* always the case for (solvable) Lie groups [51], it turns out that this right-invariant measure is also equal to the left-invariant measure,
(80)(dμ(x)=dμ(gx),)
and this left invariance will turn out to be important for the interpretation of the SPQM process as a diffusion, as will be seen in the very next section. Comments about the existence and uniqueness of the Haar measure will be given in Appendix D and Appendix E, but first it is worth taking it for granted and appreciating what can be done with it.

With the Haar measure, we can introduce the accompanying singular distributions or “*Dirac deltas*” defined by the property (sometimes called reproduction [4,73]) that, for any function *f* of G=IWH,
(81)(∫Gdμ(x)δ(y,x)f(x)=f(y).)
From the invariance properties of the Haar distribution, the Dirac delta distributions inherit the corresponding covariance properties,
(82)(δ(xg,yg)=δ(x,y)=δ(gx,gy).)
With the Haar measure and the Dirac delta, we can define a universal instrument by adding up all of the Wiener paths that end at the same Kraus operator, starting from the origin. This becomes visible by considering the total operation,
(83)ZT=∫Dμdw[0,T)O·Ldw[0,T)(84)=∫Dμdw[0,T)O·Rγdw[0,T)(85)=∫Dμdw[0,T)O·Rγdw[0,T)∫Gdμ(x)δx,γdw[0,T)(86)=∫Gdμ(x)O·R(x)∫Dμdw[0,T)δx,γdw[0,T).
In summary, the SPQM process defines a *universal instrument*, which unravels the total operation,
(87)(ZT=∫Gdμ(x)DT(x)O·R(x),)
according to a Haar-based *Kraus-operator distribution function* (KOD),
(88)(DT(x)≡∫Dμdw[0,T)δx,γdw[0,T),)
which is defined by a Wiener path integral [10,11,12,15,16,19]. The total operation is a completely positive, trace-preserving superoperator. The trace-preserving property is equivalent to saying that the POVM elements, $d\mu \left(x\right)\;D_{T}\left(x\right)\;R{\left(x\right)}^{†}R\left(x\right)$, satisfy a completeness relation,
(89)$$\begin{array}{c} 1_{H}=\int_{G}\;d\mu \left(x\right)\;D_{T}\left(x\right)\;R{\left(x\right)}^{†}R\left(x\right)\;. \end{array}$$

The term “universal” refers to the fact that this description of the instrument is common to every representation and comes from the concepts of a universal covering group and universal enveloping algebra [40,71,74]. It is important to understand that the universal covering group, IWH, is defined purely by the local structure (that is, the Lie algebra) of the observables and the quadratic term, here $H_{o}$, which accompanies the observables due to the nature of differential positive transformations. In particular, this means G=IWH is not defined by the Hilbert space of states. This ability to describe the measuring instrument without reference to a Hilbert space is so striking that we give it a name: *universal instrument autonomy* [37,40]. SPQM is in a very special class of universal instruments for which the universal covering group is finite-dimensional; we dub such instruments *principal instruments* [40].

### 2.4. Diffusion Equation in Terms of Right-Invariant Derivatives

The definition of the Kraus-operator distribution function given in Equation (Equation 88) can be thought of as a Feynman-Kac formula [15,16,19] for the solution of a Fokker-Planck-Kolmogorov (FPK) diffusion equation [59,60]. This FPK diffusion equation can be obtained easily with the help of the so-called right-invariant derivatives [39,40,46,47,48],
(90)(X←[f](x)≡limh→0f(eXhx)−f(x)h,)
which can be seen to have commutators [40]
(91)$$\underset{\leftarrow }{X}\underset{\leftarrow }{Y}-\underset{\leftarrow }{Y}\underset{\leftarrow }{X}=-\underset{\leftarrow }{\left[X,Y\right]}\;.$$
This negative sign is why left-invariant derivatives are usually considered instead of right-invariant ones. Nevertheless, because the convention is to consider operators as acting to the right, we are more-or-less stuck with having to consider a right-invariant basis for local transformations.

With the right-invariant derivatives, $D_{t+dt}$ can then be expanded about *t* in the standard way. We start with
(92)Dt+dt(x)=∫Dμdw[0,t+dt)δx,γdw[0,t+dt)(93)=∫dμ(dwt)∫Dμdw[0,t)δx,γ(dwt)γdw[0,t),
where
(94)$$\begin{array}{c} \gamma \left(dw_{t}\right)=e^{\delta_{t}}\;,\;\delta_{t}\equiv -H_{o}2\kappa \;dt+Q\sqrt{\kappa }\;dW^{q}+P\sqrt{\kappa }\;dW^{p}=-H_{o}2\kappa \;dt+a\sqrt{\kappa }\;dw_{t}^{*}+a^{†}\sqrt{\kappa }\;dw_{t} \end{array}$$
is the purely group-theoretic version of the differential Kraus operator $L(dw_{t})$ given in Equation (Equation 48); that is, $L\left(dw_{t}\right)=R\left(\gamma (dw_{t})\right)$. Here we also define the *forward generator*$\delta_{t}$: *$\gamma \left(dw_{t}\right)=e^{\delta_{t}}$ is the fundamental differential positive operator for SPQM, and the forward generator $\delta_{t}$ is thus the core mathematical object for the theory of the SPQM instrument*. Continuing with Equation (93), we have
(95)Dt+dt(x)=∫dμ(dwt)∫Dμdw[0,t)δγ(dwt)−1x,γdw[0,t)(96)=∫dμ(dwt)Dtγ(dwt)−1x(97)=∫dμ(dwt)Dte−δtx(98)=∫dμ(dwt)e−δt←[Dt]x(99)=∫dμ(dwt)Dt(x)−δt←[Dt](x)+12δt←δt←[Dt](x)(100)=Dt(x)+κdtΔ[Dt](x),
where
(101)(Δ≡2Ho←+12Q←Q←+P←P←)
is the *FPK forward generator*. Equation (Equation 95) is where the left invariance of the Haar measure is used. Equation (96) is the analog of a Chapman-Kolmogorov equation for the distribution function [59,60]. Equation (98) moves $e^{\delta_{t}}$ outside the argument of the KOD to become an exponential of the right-invariant derivative
(102)(δt←=−Ho←2κdt+Q←κdWtq+P←κdWtp,)
which we call the *vector-valued SPQM increment*. Equation (99) Taylor-expands the distribution function to the second order—that is, to order dt—as required by the Itô rule for the Wiener outcome increments. The remaining step to the FPK forward generator Δ involves averaging over the Wiener distribution $d\mu (dw_{t})$; in this averaging, the deterministic term $-2\kappa \;dt\underset{\leftarrow }{H_{o}}$ in $\underset{\leftarrow }{\delta_{t}}$ contributes a first-derivative term to the FPK forward generator, whereas the Wiener outcome increment terms in $\underset{\leftarrow }{\delta_{t}}$ contribute second-derivative diffusion terms.

In summary, the KOD of SPQM evolves according to the FPK diffusion equation,
(103)(1κ∂∂tDt(x)=Δ[Dt](x),)
with initial condition
(104)$$D_{0}\left(x\right)=\delta \left(x,1\right)\;,$$
where the group identity is defined
(105)$$1\equiv e^{0}\;,$$
which is the origin of the IWH group. Equation (Equation 103) will be (mostly) solved in Section 3.5 after having established two coordinate systems. The subtlest thing about Equation (Equation 103) is remembering that the Lie algebra of the three directions apparent in the equation means that the motion beyond the first order is actually seven-dimensional.

Before proceeding to SDEs, we draw attention to an important property of right-invariant derivatives. The reader might already be thinking about this property by wondering why we did not write $\underset{\leftarrow }{\delta_{t}}$ and Δ in terms of the right-invariant derivatives associated with *a* and $a^{†}$. The reason is that the map from operators to right-invariant derivatives, $X⟼\underset{\leftarrow }{X}$, is only R-linear and not C-linear [40]. This means that $\underset{\leftarrow }{Q}$, $\underset{\leftarrow }{P}$, $\underset{\leftarrow }{iQ}$, and $-\underset{\leftarrow }{iP}$ are R-linearly independent, with $\underset{\leftarrow }{Q}$ and $\underset{\leftarrow }{P}$ displacing Kraus operators in positive directions in the IWH group and $\underset{\leftarrow }{iQ}$ and $-\underset{\leftarrow }{iP}$ displacing in unitary directions. The right-invariant derivatives $\underset{\leftarrow }{a}$, $\underset{\leftarrow }{a^{†}}$, $-\underset{\leftarrow }{ia}$, and $\underset{\leftarrow }{ia^{†}}$ each represent a different way of equally combining displacements in the positive and unitary directions. In view of this, it is instructive to note that the vector-valued SPQM increment of Equation (Equation 102) has the form
(106)$$\begin{array}{c} \underset{\leftarrow }{\delta_{t}}=-\underset{\leftarrow }{H_{o}}2\kappa \;dt+\frac{1}{2}\sqrt{\kappa }\;dw_{t}\left(\underset{\leftarrow }{a}+\underset{\leftarrow }{a^{†}}+i\;\underset{\leftarrow }{ia}-i\;\underset{\leftarrow }{ia^{†}}\right)+\frac{1}{2}\sqrt{\kappa }\;dw_{t}^{*}\left(\underset{\leftarrow }{a}+\underset{\leftarrow }{a^{†}}-i\;\underset{\leftarrow }{ia}+i\;\underset{\leftarrow }{ia^{†}}\right)\;. \end{array}$$
This means, in particular, that
(107)$$\begin{array}{c} \Delta \neq 2\underset{\leftarrow }{H_{o}}+\frac{1}{2}\left(\underset{\leftarrow }{a}\;\underset{\leftarrow }{a^{†}}+\underset{\leftarrow }{a^{†}}\underset{\leftarrow }{a}\right)\;. \end{array}$$

### 2.5. Sample-Path SDEs in Terms of Right-Invariant One-Forms

As has been mentioned, the time-ordered exponential of the SPQM process, given in Equation (Equation 57), can be interpreted as defining sample paths in the seven-dimensional manifold G=IWH. Sample paths are usually described by stochastic differential equations (SDEs). We finish this section by explaining how such SDEs can be expressed in terms of the right-invariant structure.

The basis of right-invariant derivatives,
(108)$$\left\{\underset{\leftarrow }{e_{\nu }}\right\}\equiv \left\{-\underset{\leftarrow }{H_{o}},\underset{\leftarrow }{Q},\underset{\leftarrow }{P},-\underset{\leftarrow }{\;iP},\underset{\leftarrow }{\;iQ},-\underset{\leftarrow }{\Omega },\underset{\leftarrow }{\;i\Omega }\right\}\;,$$
defines a dual basis of right-invariant one-forms,
(109)(θμeν←≡δνμ,)
In terms of the right-invariant one-forms, the Haar measure has a simple expression in terms of wedge products,
(110)$$d\mu \left(x\right)=\theta^{-H_{o}}\wedge \theta^{Q}\wedge \theta^{P}\wedge \theta^{iQ}\wedge \theta^{-iP}\wedge \theta^{-\Omega }\wedge \theta^{i\Omega }.$$
Also, in terms of the right-invariant one-forms, the SDEs equivalent to Equation (Equation 57) are obtained by reading off the coefficient conjugate to the corresponding generator in the vector-valued SPQM increment of Equation (Equation 102),
(111)θiΩδt←=0,(112)θ−Ωδt←=0,(113)θ−iPδt←=0,(114)θiQδt←=0,(115)θQδt←=κdWtq,(116)θPδt←=κdWtp,(117)θ−Hoδt←=2κdt.

These SDEs can be broken into two types: the first-order SDEs,
(118)(θ−Hoδt←=2κdt,θQδt←=κdWtq,θPδt←=κdWtp,)
and the Pfaffians,
(119)(θ−iPδt←=θiQδt←=θ−Ωδt←=θiΩδt←=0.)
These equations will be solved in Section 3.4 after having established a coordinate system.

The SDEs of Equations (Equation 118) and (Equation 119) are almost obvious by definition, but there is a subtlety that requires attention. The right-invariant derivatives and one-forms live in the spaces tangent and cotangent to the group manifold *G* and thus are based on linear transformations that use the chain rule of ordinary calculus. Hence, the stochastic equations that come from the right-invariant one-forms are Stratonovich-form SDEs [59,60]—this means mid-point evaluation of coefficients of stochastic increments—and should be converted to the Itô-form SDEs in which coefficients are evaluated at the beginning of the increment. In the context of the IWH group, the only place this subtlety makes a difference is in the SDEs that come from $\theta^{i\Omega }$ and $\theta^{-\Omega }$. Jackson and Caves [39,40] introduced the *modified Maurer-Cartan stochastic differential* (MMCSD) as a way to get to the Itô-form equations directly. The MMCSD is an example of the Itô correction in SDEs [59,60], specifically, the Itô correction that arises when one transforms between a stochastic variable and the exponential function of that variable, as occurs in the forward generator of Equation (Equation 94). In this paper, we get directly to Itô-form SDEs in a different way when we consider the Harish-Chandra decomposition in Section 3.2.

A further subtlety about the right-invariant one-forms is that they have “curl” in the same sense as Gibbs would have defined. In the standard language of forms, this is because the exterior algebra of forms is equivalent to the Lie algebra of derivatives [46],
(120)$$\left[\underset{\leftarrow }{e_{\mu }},\underset{\leftarrow }{e_{\nu }}\right]=-{c_{\mu \nu }}^{\lambda }\underset{\leftarrow }{e_{\lambda }}\;⟺\;d\theta^{\lambda }=\frac{1}{2}{c_{\mu \nu }}^{\lambda }\;\theta^{\mu }\;\wedge \theta^{\nu }\;.$$
This equivalence is standard in modern differential topology, but an introduction is included in Appendix A; though no use will be made of the “curls” $d\theta^{\lambda }$ in this article, they are given for completeness in Equations (A15)–(A21).

## 3. The IWH Group and Two Coordinate Systems

Having introduced the instrumental Weyl-Heisenberg group, G=IWH, a coordinate system needs to be established so that we can locate the sample paths of the SPQM process and follow their propagation. If the concept of a universal covering group introduced in the previous section is unclear, seeing how Equation (Equation 57) is equivalent to a set of coordinate SDEs should help the reader appreciate that G=IWH and the SPQM process are independent of matrix representation.

We will use two coordinate systems analogous to what are called Harish-Chandra [52,53,54] (also known as “Gauss” [2,51,55]) decompositions and Cartan decompositions [50,51,52,54]. These decompositions were originally designed in the context of semisimple Lie groups [49,51,52,75,76], of which the IWH group is quite the opposite (in the sense of the Levi-Malcev decomposition). In spite of this distinction, it is very useful to think of the IWH group in many ways *as if* it were semisimple. The Harish-Chandra decomposition is easier to prove first and will allow us to make connections between the SPQM process and two processes more familiar to physicists, the Ornstein-Uhlenbeck process and the Goetsch-Graham-Wiseman (GGW) heterodyne measuring process. The Cartan decomposition is better suited for considering the POVM.

Section 3.1 identifies the analogs of the various elements of semisimple group theory. Section 3.2 proves the Harish-Chandra decomposition of the IWH group in a way that also produces the corresponding Itô-form coordinate SDEs, which are immediately recognized and solved. Section 3.3 introduces the Cartan decomposition and the transformations to Harish-Chandra coordinates and presents the right-invariant derivatives and one-forms in both coordinate systems. Section 3.4 solves the SDEs of the SPQM process in both Cartan and Harish-Chandra coordinates. Section 3.5 solves most of the FPK diffusion equation from the SPQM process in Cartan coordinates, by which we mean introducing the Cartan-section reduced distribution function and solving for it. Section 3.6 explains how the solution of the FPK diffusion equation means that the POVM of the SPQM process offers an alternative perspective on the meaning of energy quantization.

### 3.1. Usual Elements of Semisimple Lie Group Theory

As introduced in the previous section, the Lie group of interest is the so-called instrumental Weyl–Heisenberg group
(121)$$\begin{array}{c} G=\mathrm{IWH}\equiv \left\{e^{-H_{o}r}e^{Qq}e^{Pp}D_{\alpha }\;e^{i\Omega \phi }e^{-\Omega \ell }\;:\;r\in R,\;q,p\in R,\;\alpha \in C,\;\phi ,\ell \in R\right\}\;. \end{array}$$
Although *G* is literally solvable, with derived series,
(122)G=IWH⊳CWH⊳Z⊳1,
and center,
(123)$$Z\equiv \left\{e^{i\Omega \phi }e^{-\Omega \ell }\;:\;\phi ,\ell \in R\right\},$$
it can be navigated in much the same way as a semisimple group. While some of the terminology [52] will be used here, the theory of semisimple groups will be more-or-less glossed over. The purpose of this section is basically to label the various subgroups that will prove to be both meaningful and useful for navigating the IWH group and, therefore, understanding SPQM. The significance of these subgroups should become apparent as they are applied.

The map from the SPQM instrument to the SPQM POVM,
(124)(π(x)=x†x,)
defines the SPQM POVM as similar to a symmetric space, albeit a non-Riemannian one, with *Cartan group involution*
(125)$$x^{\iota }\equiv x^{-†}={\left(x^{\;-\;1}\right)}^{†}={\left(x^{†}\right)}^{\;-\;1}\;.$$

The subgroup of transformations that are even under the Cartan involution is the usual unitary Weyl-Heisenberg group of Equation (Equation 44):(126)(K≡x∈G:xι=x=π−1(1)=UWH.)
On the other hand, the remainder of *G* displaces from the origin of the *symmetric space*,
(127)(E≡π(G)≅K\G.)
Considering the conjugation action of *K* on E, almost all of the *K*-conjugacy classes can be parameterized by the *Cartan subgroup*,
(128)(A≡e−Hore−Ωℓ:r,ℓ∈R,)
and regular POVM elements are invariant under the *commutant*,
(129)(M≡k∈K:∀a∈A,kak−1=a=eiΩϕ:ϕ∈R.)
Thus the *K*-conjugacy classes have the topology of the familiar phase space,
(130)K/M≅C.
Indeed, it is the “almost all” feature where *G* and E depart from semisimple groups and Riemannian symmetric spaces, since positive transformations of the form $e^{Qq+Pp}$ are characteristically not in the conjugacy classes of *A* (see Appendix B for an additional perspective). Finally, important also are the *maximal nilpotent (or unipotent) subgroup*,
(131)(N≡eaμ*:μ∈C,)
and, perhaps the most important, the *Borel subgroup*,
(132)(B≡A⋉N=e−Hore−Ωℓeaμ*:r,ℓ∈R,μ∈C.)

This group-theoretic context now in hand, we stress that the most important groups for what follows are G=IWH itself, the center *Z* of *G*, and the quotient group G/Z=IWH/Z. The center *Z* contains a phase, which is of no importance, and a normalization, which is the main source of difficulty in analyzing SPQM. The cosets Zx∈G/Z are parametrized by what we call the ruler, *r*, and by two complex phase-space parameters. One of these complex phase-space parameters is associated with the POVM, and the other parametrizes a post-measurement displacement operator. We call G/Z the *reduced instrumental Weyl-Heisenberg* (RIWH) group. It is isomorphic to the adjoint group of *G*, but given the way we will use multipliers on the cosets Zx, we prefer to think of *G* as a central extension of G/Z.

### 3.2. Harish-Chandra Decomposition and SDEs as a Proof by Transfinite Induction

The seven-dimensional instrumental Weyl-Heisenberg group *G* affords a *Harish-Chandra decomposition*, $G=N^{†}\;MAN$, where every element can be decomposed into the form
(133)(x=ea†νe−Hor+Ωzeaμ*,)
defining seven *Harish-Chandra coordinates* (ν,r,z,μ)∈C×R×C×C. We often break the complex coordinates into real and imaginary parts,
(134)$$\nu =\frac{\nu_{1}+i\nu_{2}}{\sqrt{2}}\;,\;\mu =\frac{\mu_{1}+i\mu_{2}}{\sqrt{2}}\;,\;z=-s+i\psi \;.$$
The coordinate *r* we call the ruler, ν and μ are the postmeasurement and POVM phase-plane coordinates, and *z* is the IWH group center coordinate.

A proof that this decomposition exists for every element of the SPQM process of Equation (Equation 57) is not difficult if we allow ourselves to apply transfinite induction: at time t=dt (the first infinitesimal increment), it is easy to see that the decomposition exists because (see Equation (Equation 94))
(135)xdt=eδ=e−Ho2κdt+a†κdw0+aκdw0*(136)=e−Ho2κdtea†κdw0+aκdw0*(137)=ea†κdw0e−Ho2κdt+Ω12κ|dw0|2eaκdw0*.
Trivially, this also means that the decomposition exists for any finite integer *n* and infinitesimal time t=ndt simply because the one-parameter subgroups commute to infinitesimal order so long as the Wiener increments are independent. Now for the transfinite step. If we assume that the decomposition holds for a finite time *t*,
(138)$$x_{t}=e^{a^{†}\nu_{t}}e^{-H_{o}r_{t}+\Omega z_{t}}e^{a\mu_{t}^{*}}\;,$$
then an increment later in the SPQM process, we have
(139)xt+dt=eδtxt(140)=ea†κdwte−Ho2κdt+Ω12κ|dwt|2eaκdwt*ea†νte−Hort+Ωzteaμt*(141)=ea†κdwte−Ho2κdtea†νteaκdwt*e−Hort+Ω(zt+12κ|dwt|2+νtκdwt*)eaμt*(142)=ea†(e−2κdtνt+κdwt)e−Ho(r+2κdt)+Ω(z+12κ|dwt|2+νtκdwt*)ea(μt+e−rtκdwt)*.
This concludes the proof of the Harish-Chandra decomposition for the SPQM process and G=IWH.

A consequence of the proof is that the SPQM process in Harish-Chandra coordinates is equivalent to the Itô-form SDEs [56,57,58,59,60],
(143)drt=2κdt,(144)dνt=−2κdtνt+κdwt,(145)dμt=e−rtκdwt,(146)−dst+idψt=dzt=12κ|dwt|2+νtκdwt*.
Although these are Itô-form SDEs, notice that we did not use the Itô rule in deriving them; in particular, we do not set $|dw_{t}{|}^{2}=dt$ in the SDE for the center coordinate *z*.

We now solve these equations for the initial condition $r_{0}=\nu_{0}=\mu_{0}=z_{0}=0$. These initial values are chosen so that $x_{0}=1$, in agreement with the δ-function initial condition for KOD, which is given in Equation (Equation 104). It is straightforward to see that the first three SDEs have as solutions
(147)(rT=2κT,)
(148)(νT=∫0T−κdwte−2κ(T−t),)
(149)(μT=∫0T−κdwte−2κt,)
where $T_{-}\equiv T-dt$. The fourth equation is solved by plugging the solution for $\nu_{t}$ into the equation for $z_{t}$ and integrating, with the result that
(150)(−sT+iψT=zT=12∫0T−∫0T−κdwt*dwse−2κ|t−s|1+sgn(t−s),)
where
(151)$$\begin{array}{c} \mathrm{sgn}\left(u\right)=\left\{\begin{array}{cc} 1, & u>0,\\ 0, & u=0,\\ -1, & u<0, \end{array}\right. \end{array}$$
is the sign function. There being subtleties in deriving and interpreting this solution, it is worked out carefully in Appendix C. The solutions for the real and imaginary parts of *z* are
(152)−sT=12∫0T−∫0T−κdwt*dwse−2κ|t−s|,(153)iψT=12∫0T−∫0T−κdwt*dwse−2κ|t−s|sgn(t−s).

The SPQM posterior phase-point variable, $\nu_{T}$ from Equation (Equation 148), is a linear functional in the measurement record and can be recognized as an Ornstein-Uhlenbeck (OU) process [59,60,61]. The SPQM prior phase-point variable, $\mu_{T}$ from Equation (Equation 149), is a linear functional in the measurement record and can be recognized as the same as the Goetsch-Graham-Wiseman (GGW) heterodyne process. (To be clear, the equations for the remaining variables of the GGW heterodyne process would instead be $r_{T}^{\mathrm{GGW}}=2\kappa T$, $\nu_{T}^{\mathrm{GGW}}=0$, and $z_{T}^{\mathrm{GGW}}=\kappa T$ [37]; in particular, the GGW heterodyne process corresponds to the Borel subgroup *B*.) The center SPQM variable, $z_{T}$ from Equation (Equation 150), is a quadratic functional in the measurement record; the solution is identical to that obtained by deriving a quantum fluctuation-dissipation theorem for the correlation of ν and μ.

It is worth taking a moment to reflect further on Equations (Equation 147)–(Equation 150). Equation (Equation 147) tells us that if $H_{o}$ is quantized in the standard way, then the Kraus operators collapse, with an *e*-folding time τ=1/2κ, to a scaled outer product of coherent states of the form
(154)$$L_{T\gg 1/\kappa }∼e^{-\kappa T+z_{T}}e^{a^{†}\nu_{T}}\left|0\rangle \;\langle 0\right|e^{a\mu_{T}^{*}}=e^{i\psi_{T}}e^{\frac{1}{2}\left(\right|\nu_{T}{|}^{2}+|\mu_{T}{|}^{2})-s_{T}-\kappa T}|\nu_{T}\rangle \langle \mu_{T}|\;.$$
The complementarity in time of the OU and GGW processes (Equations (Equation 148) and (Equation 149)) is also interesting: whereas the post-measurement variable $\nu_{T}$ of Equation (Equation 148) depends only on the end of the outcome register, the POVM variable $\mu_{T}$ of Equation (Equation 149) depends only on the beginning of the register. It is thus reasonably clear that the POVM of the SPQM process culminates in the usual “measurement in the coherent-state basis”,
(155)$$1_{H}=\int \frac{d^{2}\mu }{\pi }|\mu \rangle \;\langle \mu |\;,$$
where $d^{2}\mu =\frac{1}{2}d\mu_{1}d\mu_{2}$. This is just like GGW heterodyne detection, except that the post-measurement state is not vacuum, but instead is scrambled over phase space. That the POVM variable $\mu_{T}$ ceases to evolve after a few *e*-foldings seems to be an important feature in the interpretation of SPQM as a measuring process [66]. All this said, there remains the elephant in the room that prompts us to label these conclusions as “reasonably clear”: the elephant is the factor that scales the long-time Kraus operator in Equation (Equation 154); although Equation (Equation 154) makes it absolutely clear that the Kraus operator approaches an outer product of coherent states, what is not clear at all is how the POVM approaches a uniform distribution of coherent states, as required by the completeness relation given in Equation (Equation 155). Figuring this out is, in some sense, what the rest of the paper is about.

### 3.3. Cartan Decomposition and Various Transformations

As far as what the SPQM process is ultimately doing in time, Section 3.2 in many ways says it all—except for that elephant in the room. We now turn our attention to a detailed understanding of the measuring process at finite times, which requires addressing the elephant. Key to this is the realization that the Harish-Chandra decomposition is not well suited to telling us the behavior of the POVM. Rather, a Cartan decomposition of the instrumental Weyl-Heisenberg group, G=KAK, works best for this purpose. A straightforward calculation (left to Appendix B) shows that almost every group element—and every Kraus operator—with a Harish-Chandra decomposition also affords a *Cartan decomposition*,
(156)(x=DβeiΩϕe−Hor−ΩℓDα−1,)
and, therefore, the POVM elements decompose as
(157)(x†x=Dαe−Ho2r−Ω2ℓDα−1,)
where the *Cartan coordinates*, (β,ϕ,r,ℓ,α)∈C×R×R×R×C, can be obtained by the coordinate transformations (the ruler *r* is shared by the two coordinate systems),
(158)β=erν+μ2sinhr,(159)α=erμ+ν2sinhr,(160)ℓ=s−f,(161)ϕ=ψ−ξ.
Here *f* and ξ are functions of the ruler and the phase-plane coordinates, that is, functions on the RIWH G/Z,
(162)f=(|ν|2+|μ|2)er+ν*μ+νμ*4sinhr=|ν+μ|24(1−e−r)+|ν−μ|24(1+e−r)=12|β|2+|α|2−β*αe−r−βα*e−r=1−e−r4|β+α|2+1+e−r4|β−α|2,(163)ξ=ν*μ−νμ*4isinhr=(ν−μ)*(ν+μ)−(ν−μ)(ν+μ)*8isinhr=e−rβ*α−βα*2i=e−r(β−α)*(β+α)−(β−α)(β+α)*4i.
Notice the singularity in Cartan coordinates at r=0, about which there is further discussion throughout the remainder of the paper and, in particular, in Appendix B.

As we wish to solve Equations (Equation 103) and (111)–(117), more relevant to our purpose are the transformations from the right-invariant moving frame to the Cartan coordinate frame. A calculation of these frame transformations can be found in Appendix D, but we include them here for continuity; we also include the transformations to the Harish-Chandra coordinate frame, which are worked out in Appendix E. For the FPK diffusion equation of Equation (Equation 103), we require the transformations of the derivatives,
(164)(iΩ←=∂ϕ=∂ψ(−Ω←=∂ℓ=∂s(−iP←=∂β1−12β2∂ϕ=∂ν1−e−r∂μ1+12ν1∂s−12ν2∂ψ(iQ←=∂β2+12β1∂ϕ=∂ν2−e−r∂μ2+12ν2∂s+12ν1∂ψ(Q←=∇1−β1∂ℓ+β2coshr−α22sinhr∂ϕ=∇1−12ν1∂s+12ν2∂ψ(P←=∇2−β2∂ℓ−β1coshr−α12sinhr∂ϕ=∇2−12ν2∂s−12ν1∂ψ(−Ho←=∂rC−β1∇1−β2∇2+β12+β222∂ℓ+β1α2−β2α12sinhr∂ϕ=∂rHC−ν1∂ν1−ν2∂ν2,
where
(165)$$\nabla_{j}\equiv \frac{1}{\sinh r}\left(\partial_{\alpha_{j}}+\cosh r\;\partial_{\beta_{j}}\right)=\partial_{\nu_{j}}+e^{-r}\partial_{\mu_{j}}\;.$$
The two coordinate systems share the coordinate *r*, but the partial derivative with respect to *r* is, of course, different in the two systems. In the above equation we distinguish $\partial_{r}$ in the two systems, but we do this nowhere else because it is always clear in which coordinate system we are operating. It is worth recording for working in terms of the complex phase-space variables that
(166)∇≡12(∇1−i∇2)=1sinhr∂α+coshr∂β=∂ν+e−r∂μ,(167)∇*=12(∇1+i∇2)=1sinhr∂α*+coshr∂β*=∂ν*+e−r∂μ*.

For the SDEs of Equations (111)–(117), we require the transformations of the one-forms,
(168)(θiΩ=dϕ+12(β2dβ1−β1dβ2)+12(α2dα1−α1dα2)+coshr(β1dα2−β2dα1)=dψ+12er(ν1dμ2−ν2dμ1)(θ−Ω=dℓ+12(β12+β22)dr+sinhr(β1dα1+β2dα2)=ds+12er(ν1dμ1+ν2dμ2)(θ−iP=dβ1−coshrdα1=12dν1−erdμ1+ν1dr(θiQ=dβ2−coshrdα2=12dν2−erdμ2+ν2dr(θQ=β1dr+sinhrdα1=12dν1+erdμ1+ν1dr(θP=β2dr+sinhrdα2=12dν2+erdμ2+ν2dr(θ−Ho=dr=dr.
With the one-form transformations in hand, the Haar measure, written in terms of the one-forms in Equation (Equation 110), becomes in the two coordinate systems,
(169)(d7μ(x)=dϕdℓd2βπdrsinh2rd2απ=dψdsd2ν2πdre2rd2μ2π.)
Here and elsewhere, complex phase-plane measures are denoted by $d^{2}\beta =\frac{1}{2}d\beta_{1}d\beta_{2}$. The factors of 1/π in the Cartan measure are conventional in quantum optics and ultimately come from the coherent-state completeness relation of Equation (Equation 155). The factors of 1/2π in Harish-Chandra coordinates then follow from the transformation from Cartan to Harish-Chandra coordinates. The left invariance of these measures is discussed in Appendix D and Appendix E. While we are considering measures, we should record the reduced measure on the five-dimensional group RIWH=IWH/Z:(170)(d5μ(Zx)=d7μ(x)dμ(Z)=d2βπdrsinh2rd2απ=d2ν2πdre2rd2μ2π.)
Here dμ(Z)=dϕdℓ=dψds is the measure on the center *Z*.

Accompanying the coordinate Haar measure $d^{7}\;\mu \left(x\right)$ are the coordinate forms of the conjugate δ-function,
(171)δ(x,x′)=δ(ϕ−ϕ′)δ(ℓ−ℓ′)1sinh2rδ(r−r′)πδ2(β−β′)πδ2(α−α′)(172)=δ(ψ−ψ′)δ(s−s′)e−2rδ(r−r′)2πδ2(ν−ν′)2πδ2(μ−μ′).
There are obvious coordinate forms for the δ-function $\delta (Zx,Zx^{\prime })$ that is conjugate to $d^{5}\;\mu \left(Zx\right)$: (173)δ(Zx,Zx′)=1sinh2rδ(r−r′)πδ2(β−β′)πδ2(α−α′)(174)=e−2rδ(r−r′)2πδ2(ν−ν′)2πδ2(μ−μ′).
We are especially interested in δ(x,1). As the identity 1 has Harish-Chandra coordinates ϕ=ℓ=r=0 and ν=μ=0, we have
(175)$$\begin{array}{c} \delta \left(x,1\right)=\delta \left(\phi \right)\;\delta \left(s\right)\;\delta \left(r\right)\;2\pi \delta^{2}\left(\nu \right)\;2\pi \delta^{2}\left(\mu \right)\;. \end{array}$$
The Cartan form of δ(x,1) requires more attention because of the coordinate singularity at r=0 (see Appendix B for discussion). The singularity is about more than just the $1/\sinh^{2}\;r$ in the Cartan form of the δ-function, although that singularity is the root of the difficulties that require attention. We provide the necessary attention in Appendix F.

### 3.4. Solving the SDEs

Section 2.5 left off by showing that SPQM corresponds to the right-invariant Stratonovich-form SDEs of Equations (111)–(117). With the frame transformations from Equation (Equation 168) of the previous section at hand, the three first-order SDEs, given in Equation (Equation 118), find the following expressions,
(176)2κdt=θ−Ho(δt←)=dr,(177)κdwt=12θQ(δt←)+iθP(δt←)=βdr+sinhrdα=12dν+erdμ+νdr,
and the four Pfaffians, displayed in Equation (Equation 119), give
(178)0=12θ−iP(δt←)+iθiQ(δt←)=dβ−coshrdα=12dν−erdμ+νdr,(179)0=θ−Ω(δt←)=dℓ+12(β12+β22)dr+sinhr(β1dα1+β2dα2)=ds+12er(ν1dμ1+ν2dμ2),(180)0=θiΩ(δt←)=dϕ+12(β2dβ1−β1dβ2)+12(α2dα1−α1dα2)+coshr(β1dα2−β2dα1)=dψ+12er(ν1dμ2−ν2dμ1),
where the evaluation of the coordinate one-forms on the vector-valued SPQM increment, $\underset{\leftarrow }{\delta_{t}}$, is no longer denoted.

Equation (Equation 176) is the ruler equation given previously in Equation (Equation 143); it has the solution $r_{t}=2\kappa t$ for the initial condition $r_{0}=0$. Summing and differencing Equations (177) and (178), one finds that
(181)$$\begin{array}{c} \begin{array}{cc} \sqrt{\kappa }\;dw_{t} & =d\beta +\beta \;dr-e^{-r}d\alpha \\ \; & =d\nu +\nu \;dr\;, \end{array} \end{array}$$
(182)$$\begin{array}{c} \begin{array}{cc} \sqrt{\kappa }\;dw_{t} & =-d\beta +\beta \;dr+e^{r}d\alpha \\ \; & =e^{r}d\mu \;. \end{array} \end{array}$$
As was discussed in Section 2.5, these are Stratonovich-form SDEs, which means that the coefficients are evaluated at the midpoint, $t+\frac{1}{2}\;dt$, of the increment—technically, the midpoint has no status in the stochastic calculus, so one can regard midpoint evaluation as $a_{t+dt/2}=\frac{1}{2}\left(a_{t}+a_{t+dt}\right)=a_{t}+\frac{1}{2}da_{t}$—but for these SDEs, that evaluation does not produce an Itô correction (because $dr_{t}=2\kappa dt$ has no stochastic term), and the equations can be read as Itô-form SDEs, with the coefficients evaluated at the beginning of the increment. When *r* and dr are substituted into these SDEs, the equations for the Harish-Chandra complex phase-point coordinates, ν and μ, become Equations (144) and (145), with the solutions shown in Equations (Equation 148) and (Equation 149) for the initial conditions $\nu_{0}=0$ and $\mu_{0}=0$.

The SDEs for Cartan phase-space points are
(183)dβ=cosh2κtdα,(184)dα=csch2κt(−2κβdt+κdwt).
For integrating, these SDEs are more profitably written as
(185)d(νe2κt)=d(βe2κt−α)=e2κtκdwt,(186)dμ=d(−βe−2κt+α)=e−2κtκdwt,
and the reason is that these are the same as integrating the SDEs for the Harish-Chandra phase-plane coordinates. The solutions, satisfying initial conditions $\nu_{0}=\mu_{0}=0$,
(187)νTe2κT=βTe2κT−αT=∫0T−κdwte2κt,(188)μT=−βTe−2κT+αT=∫0T−κdwte−2κt,
are those displayed in Equations (Equation 148) and (Equation 149) for the Harish-Chandra phase-space points. Summarizing, we have that the solutions for the ruler and the Cartan phase-space coordinates are
(189)(rT=2κT,βT=∫0T−κdwtcosh2κtsinh2κT,αT=∫0T−κdwtcosh2κ(T−t)sinh2κT.)

The Stratonovich-form SDE for the Harish-Chandra center coordinate *z* follows from the SDEs of Equations (179) and (180),
(190)$$\begin{array}{c} dz_{t}=-ds_{t}+id\psi_{t}=e^{r_{t+dt/2}}\nu_{t+dt/2}d\mu_{t}^{*}\;. \end{array}$$
Converted to Itô form, this equation is
(191)$$\begin{array}{c} \begin{array}{cc} dz_{t} & =(\nu_{t}+\frac{1}{2}d\nu_{t})\;e^{r_{t}}\;d\mu_{t}^{*}\\ \; & =\left(\nu_{t}+\frac{1}{2}\sqrt{\kappa }\;dw_{t}\right)\sqrt{\kappa }dw_{t}^{*}\\ \; & =\frac{1}{2}\kappa {\left|dw_{t}\right|}^{2}+\nu_{t}\;dw_{t}^{*}\;, \end{array} \end{array}$$
in agreement with Equation (146). The solution for the initial condition $z_{0}=0$ is carefully worked out in Appendix C and given in Equation (Equation 150).

The Cartan normalization and phase-space coordinates have Stratonovich-form SDEs,
(192)−dℓ=12(β12+β22)dr+sinhr(β1dα1+β2dα2),(193)dϕ=12(β1dβ2−β2dβ1)+12(α1dα2−α2dα1)+coshr(β2dα1−β1dα2),
where all the coefficients are evaluated at the midpoint $t+\frac{1}{2}dt$. Converted to equivalent Itô-form SDEs, these equations become
(194)−dℓt=|β|2dr+sinhr(βdα*+β*dα)+12sinhr(dβdα*+dβ*dα)(195)=coth2κtκ|dwt|2−2|βt|2κdt+βtκdwt*+βt*κdwt,(196)idϕ=12(β*dβ−βdβ*+α*dα−αdα*)+coshr(βdα*−β*dα)+12coshr(dβdα*−dβ*dα)(197)=csch2κt(αtβt*−αt*βt)κdt+12(βtcoth2κt−αtcsch2κt)κdwt*−12(βt*coth2κt−αt*csch2κt)κdwt.
Though more complicated, these equations have a similar character to the SDEs for the Harish-Chandra center coordinate z=−s+iψ. The Itô correction for dℓ is the last term in Equation (Equation 194), and it becomes the coth term at the beginning of Equation (195), whereas the Itô correction for dϕ at the end of Equation (196) vanishes. We could solve these equations directly, but it is both easier and more productive to combine the solution for *z* with the coordinate transformation to Cartan coordinates, thus giving
(198)(ℓT=sT−fTandϕT=ψT−ξT,)
where $s_{T}$ and $\psi_{T}$ are the Harish-Chandra solutions shown in Equations (152) and (153) and $f_{T}$ and $\xi_{T}$ are the functions from Equations (162) and (163) with all the coordinates evaluated at time *T*.

### 3.5. Solving Most of the FPK Diffusion Equation

Section 2.4 left off showing that the sample paths of SPQM diffuse according to the KOD $D_{t}\left(x\right)$, which satisfies the FPK equation given in Equation (Equation 103), with the initial condition $D_{0}\left(x\right)=\delta \left(x,1\right)$. The crucial mathematical object in the diffusion equation is the FPK forward generator Δ, which is written in terms of right-invariant derivatives in Equation (Equation 101).

With the frame transformations of Equation (Equation 164) at hand, it is easy to express the three pieces of the FPK forward generator in Cartan coordinates,
(199)$$\begin{array}{c} 2\underset{\leftarrow }{H_{o}}=-2\partial_{r}+2\left(\beta_{1}\nabla_{1}+\beta_{2}\nabla_{2}-\frac{\beta_{1}^{2}+\beta_{2}^{2}}{2}\partial_{\ell }\right)+\partial_{\phi }B_{H_{o}}\;, \end{array}$$
(200)12Q←Q←=12∇1−β1∂ℓ+β2coshr−α22sinhr∂ϕ2(201)=12∇1−β1∂ℓ2+12∂ϕBQ(202)=12∇12−∇1β1+β1∇1∂ℓ+β12∂ℓ2+12∂ϕBQ(203)=12∇12−cothr+2β1∇1∂ℓ+β12∂ℓ2+12∂ϕBQ(204)=−12cothr∂ℓ+12∇12−∂ℓβ1∇1−12β12∂ℓ+12∂ϕBQ,(205)12P←P←=12∇2−β2∂ℓ−β1coshr−α12sinhr∂ϕ2(206)=12∇2−β2∂ℓ2+12∂ϕBP(207)=−12cothr∂ℓ+12∇22−∂ℓβ2∇2−12β22∂ℓ+12∂ϕBP,
where we introduce the quantities
(208)BHo=β2α1−β1α2sinhr,(209)BQ=β2coshr−α2sinhr∇1−β1∂ℓ+β2coshr−α22sinhr2∂ϕ,(210)BP=−β1coshr−α1sinhr∇2−β2∂ℓ+β1coshr−α12sinhr2∂ϕ,
which are independent of ϕ and commute with $\partial_{\phi }$. The ϕ-derivative terms quickly disappear from the analysis, ultimately because ϕ is irrelevant to the instrument elements as a consequence of the symmetry $O\;\cdot \;\left(e^{i\Omega \phi }L\right)=O\;\cdot \;\left(L\right)$. Putting these expressions together, the FPK forward generator of Equation (Equation 101) becomes in Cartan coordinates,
(211)(Δ=−2∂r−cothr∂ℓ+12∇12+∇22+(2−∂ℓ)β1∇1+β2∇2−β12+β222∂ℓ+∂ϕB,)
where $B=B_{H_{o}}+\frac{1}{2}B_{Q}+\frac{1}{2}B_{P}$. It is worth noting that
(212)$$\begin{array}{c} \frac{1}{2}\left(\nabla_{1}^{2}+\nabla_{2}^{2}\right)=\nabla^{*}\nabla =\nabla \nabla^{*}\;, \end{array}$$
where ∇ and $\nabla^{*}$, defined in Equations (166) and (167), are derivatives with respect to complex phase-space coordinates.

Due to the cubic and quartic nature of the last few terms in Equation (Equation 211), we do not hope to find a complete analytic solution to Equation (Equation 103). However, “5/7-ths” of the distribution can be analyzed quite easily. Remember that we are interested in the instrument, and observe that the instrument elements can be partitioned by reconsidering the total operation $Z_{T}$, written in terms of $D_{T}\left(x\right)$ in Equation (Equation 87), as
(213)ZT=∫Gd7μ(x)DT(x)O·(x)(214)=∫−1.5ptG/Zd5μ(Zx)∫ZdϕdℓDT(x)e−2ℓO·Dβe−HorDα†(215)=∫−1.5ptG/Zd5μ(Zx)CT(Zx)O·Dβe−HorDα†,
where we use the coset measure $d^{5}\;\mu \left(Zx\right)$ introduced in Equation (Equation 170) and define the *Cartan-section reduced distribution function*,
(216)(CT(Zx)≡∫ZdϕdℓDT(x)e−2ℓ.)

Readers uncomfortable with the coset notation can think that, in this equation, $x=e^{i\Omega \phi }e^{-\Omega \ell }D_{\beta }\;e^{-H_{o}r}D_{\alpha }^{†}$ and $Zx=D_{\beta }\;e^{-H_{o}r}D_{\alpha }^{†}$. Even more prosaically, one can regard $D_{T}$ as being a function of all seven Cartan coordinates and $C_{T}$ as being a function of five of them, the ruler *r* and the complex phase-space coordinates β and α. Our excuse—quite a good excuse, really—for using the coordinate-independent coset notation is that we will elaborate on this distribution function and another one in Section 4, but working there mainly in Harish-Chandra coordinates.

Equation (215) is a new unraveling of the SPQM instrument, which we call the *reduced SPQM instrument*, with instrument elements
(217)$$\begin{array}{c} d^{5}\;\mu \left(Zx\right)\;C_{T}\left(Zx\right)\;O\;\cdot \;\left(D_{\beta }\;e^{-H_{o}r}D_{\alpha }^{†}\right)\;, \end{array}$$
in which the Cartan reduced distribution $C_{T}\left(Zx\right)$ is conjugate to the operation $O\;\cdot \;(D_{\beta }\;e^{-H_{o}r}D_{\alpha }^{†})$. If $H_{o}$ is quantized in the standard way, we have, at late times,
(218)$$\begin{array}{c} D_{\beta }\;e^{-H_{o}r}D_{\alpha }^{†}{|}_{T\gg 1/\kappa }∼e^{-\kappa T}D_{\beta_{T}}\left|0\rangle \;\langle 0\right|D_{\alpha_{T}}^{†}=e^{-\kappa T}|\beta_{T}\rangle \langle \alpha_{T}|\;. \end{array}$$
This makes clear that $C_{T}\left(Zx\right)$ is the elephant in the room hinted at in Section 3.2: it is the phase-space weighting function that is crucial for the POVM completeness relation, which takes the form
(219)$$\begin{array}{c} 1_{H}=\int_{G}\;d^{7}\;\mu \left(x\right)\;D_{T}\left(x\right)\;x^{†}x=\int_{-1.5ptG/Z}\;\;\;\;\;d^{5}\;\mu \left(Zx\right)\;C_{T}\left(Zx\right)\;D_{\alpha }^{†}e^{-H_{o}2r}D_{\alpha }\;. \end{array}$$

As an elephant, however, $C_{t}\left(Zx\right)$ is not normalized to unity—indeed, its normalization is ill defined. Moreover, $C_{t}\left(Zx\right)$ is not the weight function whose moments are those of the Cartan phase-point variables β and α according to the SDE solutions given in Equation (Equation 189). The distribution function that does give these moments is the straight marginal of $D_{T}\left(x\right)$ over the center *Z*,
(220)$$D_{T}\left(Zx\right)\equiv \int_{Z}d\phi \;d\ell \;D_{T}\left(x\right)\;.$$
This distribution is normalized and has finite moments, those coming from the SDE solutions of Equation (Equation 189). For those very reasons, however, $D_{T}\left(Zx\right)$ cannot possibly give rise to a POVM completeness relation; it will not be seen again in this paper.

To derive an evolution equation for the reduced distribution function $C_{t}\left(Zx\right)$ of Equation (Equation 216), one takes its time derivative, substitutes omtp tje omtegra; the FPK equation given in Equation (Equation 103), and pushes the FPK forward generator Δ from Equation (Equation 211) through the center integrals by integrating by parts. Integration by parts on ϕ gets rid of the derivatives with respect to ϕ, and integration by parts on *ℓ* translates to substituting $\partial_{\ell }\to 2$, resulting in the partial differential equation (PDE),
(221)(1κ∂∂tCt(Zx)=−2∂r−2cothr+∇*∇[Ct](Zx).)
for which one should recall that $\nabla^{*}\nabla =\frac{1}{2}\left(\nabla_{1}^{2}+\nabla_{2}^{2}\right)$. This PDE is ballistic in the ruler *r*—solution proportional to δ(r−2κt)—and Gaussian-preserving in the phase-space variables, but as a consequence of the −2cothr term, the PDE does not preserve normalization [15,16,19].

To solve for $C_{t}\left(Zx\right)$ requires knowing $D_{dt}\left(x\right)$, which is, when dt→0, the δ-function δ(x,1). This can be done fairly easily by evaluating the Cartan-coordinate solutions from Equations (Equation 189) and (Equation 198) at T=dt. We perform that task in Appendix F, where we also identify all the δ-function forms for initial conditions. The result for $D_{dt}\left(x\right)$ is
(222)$$\begin{array}{c} D_{dt}\left(x\right)=\delta \left(\phi \right)\;\delta \left(\ell +\frac{1}{2}\kappa \;dt{\left|\beta +\alpha \right|}^{2}\right)\;\frac{1}{\sinh^{2}\;r}\delta \left(r-2\kappa \;dt\right)\;2\pi \frac{\kappa \;dt}{\pi }e^{{-\kappa \;dt|\beta +\alpha |}^{2}}\;2\pi \delta^{2}\left(\beta -\alpha \right)\;. \end{array}$$
The distinctive feature of this distribution is the wide, normalized Gaussian in β+α, which limits to a uniform distribution in β+α as dt→0. What the wide Gaussian is about is the fact that the identity is represented by Cartan coordinates ϕ=ℓ=r=0 and β=α, with β+α free to take on any complex value. With this expression, it is easy to see that
(223)(Cdt(Zx)=∫ZdϕdℓDdt(x)e−2ℓ=2rδ(r−2κdt)πδ2(β−α).)
The integral of this distribution has a zero from the 1/r behavior multiplying the $\sinh^{2}r$ in the measure $d^{5}\;\mu \left(Zx\right)$ and an infinity from the uniformity in β+α; therefore, the normalization is ill defined. This ill-defined normalization is, however, exactly what is needed to give a well-defined POVM. For these reasons, $C_{dt}\left(Zx\right)$ does not limit to δ(Zx,Z1) as dt→0, for which see Appendix F.

The initial condition $C_{dt}\left(Zx\right)$ is independent of β+α, and the distribution $C_{t}\left(Zx\right)$ remains so under the PDE of Equation (Equation 221). To see the consequences most clearly, it is useful to transform to sum and difference variables,
(224)$$\begin{array}{c} \beta_{\pm }=\beta \pm \alpha \;, \end{array}$$
in which the covariant derivative ∇ of Equation (Equation 166) becomes
(225)$$\begin{array}{c} \nabla =\coth\left(r/2\right)\partial_{\beta_{+}}+\tanh\left(r/2\right)\partial_{\beta_{-}}\;. \end{array}$$
Therefore, the weight function evolves according to the PDE,
(226)$$\frac{\partial }{\partial t}C_{t}\left(Zx\right)=\kappa \left(-2\coth r-2\partial_{r}+\tanh^{2}\left(r/2\right)\partial_{\beta_{-}^{*}}\partial_{\beta_{-}}\right)C_{t}\left(Zx\right)\;,$$
with solution
(227)(CT(Zx)=1sinhrδ(r−2κT)2π1πΣTe−|β−α|2/ΣT.)
The width of the difference in phase points, $\Sigma_{T}$, satisfies the differential equation, $d\Sigma_{t}/dt=\kappa \tanh^{2}\;\kappa t$, with a solution, given the initial condition $\Sigma_{0}=0$,
(228)(ΣT=κT−tanhκT.)

In summary, the SPQM instrument can be considered as the *reduced SPQM instrument unraveling*,
(229)(ZT=∫d5μ(Zx)CT(Zx)O·Dβe−HorDα†,)
with instrument elements
(230)(d5μ(Zx)CT(Zx)O·(Dβe−HorDα†)=2sinh2κTdrδ(r−2κT)d2απd2βπΣTe−|β−α|2/ΣTO·(Dβe−HorDα†),)
where the width $\Sigma_{T}$ of the difference in phase points is given by Equation (Equation 228). There are four notable features in the temporal behavior of the reduced SPQM instrument:The ruler *r* (or purity parameter) is ballistic, which means that $e^{-H_{o}r}$ collapses exponentially to $e^{-\kappa T}\left|0\rangle \;\langle 0\right|$ in the standard quantization. More generally, $D_{\beta }\;e^{-rH_{o}}D_{\alpha }^{†}$ collapses exponentially at late times to an outer product of coherent states, $e^{-\kappa T}\left|\beta \rangle \langle \alpha \right|$;The dependence on the future and past phase-space parameters, β and α, is only in their difference;The distribution of the difference spreads out very slowly for small times as $\Sigma_{T}\propto T^{3}$ and then for long times becomes normal diffusion, with $\Sigma_{T}\propto T$;There is a center normalization, 2sinh2κT, that increases over time.
This center normalization is the focus of the next section, which finds that the elephant provides an alternative perspective on the quantum.

### 3.6. POVM as an Alternative Perspective on the Quantum

The center normalization just mentioned is remarkable in that it is equivalent to traditional energy quantization. Specifically, the completeness relation for the SPQM process, shown in Equation (Equation 219), is
(231)$$\begin{array}{c} 1_{H}=2\sinh2\kappa T\int \frac{d^{2}\alpha }{\pi }D_{\alpha }\;e^{-H_{o}4\kappa T}D_{\alpha }^{†}\int \frac{d^{2}\beta }{\pi \Sigma_{T}}e^{{-|\beta -\alpha |}^{2}/\Sigma_{T}}=2\sinh2\kappa T\int \frac{d^{2}\alpha }{\pi }D_{\alpha }\;e^{-H_{o}4\kappa T}D_{\alpha }^{†}\;. \end{array}$$
It is important to appreciate that, for late times T≫1/κ, when $e^{-H_{o}4\kappa T}$ collapses to $e^{-2\kappa T}\left|0\rangle \;\langle 0\right|$ in the standard quantization, this completeness relation becomes the coherent-state resolution of the identity of Equation (Equation 155):(232)$$\begin{array}{c} 1_{H}=\int \frac{d^{2}\alpha }{\pi }|\alpha \rangle \;\langle \alpha |\;. \end{array}$$

The completeness relation says much more, however, when considered for arbitrary times *T*. If H is an irreducible representation, Schur’s lemma says that
(233)$$\begin{array}{c} \int \frac{d^{2}\alpha }{\pi }D_{\alpha }\;e^{-H_{o}4\kappa T}D_{\alpha }^{†}=1_{H}\mathrm{tr}\;\left(e^{-H_{o}4\kappa T}\right)\;. \end{array}$$
The trace, which one recognizes as a partition function, is defined within the representation and is evaluated using traditional energy quantization as
(234)$$\begin{array}{c} \mathrm{tr}\left(e^{-H_{o}4\kappa T}\right)=\sum_{n=0}^{\infty }e^{-(n+\frac{1}{2})4\kappa T}=\frac{1}{2\sinh2\kappa T}\;. \end{array}$$
The center normalization in the POVM completeness relation thus evaluates the partition function of $e^{-H_{o}4\kappa T}$ without using traditional energy quantization. This is to be expected in view of the Stone-von Neumann theorem, but expected though it is, please appreciate how different the setting of this paper is from the original ideas of energy quantization and thermal equilibrium. Remember that here the operator $H_{o}$ comes from the trace-preserving character of the instrument. It is not the energy of the system; indeed we have explicitly eschewed any notion of system energy or of a Hamiltonian. The operator $H_{o}$ plays the role of a “dissipator”—specifically, a dissipator that damps the POVM to the coherent states—but there is no notion of energy associated with this dissipation. The coherent states and $H_{o}$ arise within a group structure constructed solely from the measured observables, *Q* and *P*. Moreover, the parameter conjugate to $H_{o}$, the ruler *r*, is quite literally time rather than an inverse temperature. It seems remarkable that this result holds, from the completeness of the SPQM POVM, without any assumption of a Hamiltonian, a ground state, or thermal equilibrium.

## 4. Reduced Distribution Functions and Feynman-Kac Path Integrals

This section further considers reduced distribution functions, their path-integral expressions and diffusion equations, and their relation to SDEs. The path-integral expression,
(235)$$D_{T}\left(x\right)=\int D\mu \;\left[dw_{\left[0,T\right)}\right]\delta \left(x,\gamma \left[dw_{\left[0,T\right)}\right]\right)\;,$$
is generally considered to be a Feynman-Kac formula [15,16,19] for the associated diffusion equation. Our analysis is rooted in the path integral of Equation (Equation 58) for the overall SPQM quantum operation,
(236)$$\begin{array}{c} Z_{T}=\int D\mu \;\left[dw_{\left[0,T\right)}\right]O\;\cdot \;\left(R\left(\gamma \left[dw_{\left[0,T\right)}\right]\right)\right)\;. \end{array}$$
Equation (Equation 236) expresses the relation between a complex Wiener path $dw_{\left[0,T\right)}$, a point in the group manifold IWH, $\gamma \left[dw_{\left[0,T\right)}\right]$, and, in turn, the overall Kraus operator $R(\gamma \left[dw_{\left[0,T\right)}\right])$ written as a time-ordered product of incremental Kraus operators. Equation (Equation 235) defines the Kraus-operator distribution function (KOD) as the amalgamation of all paths that lead to the same Kraus operator. The KOD inherits the path-integral expression given in Equation (Equation 236), and from this path integral, one can derive a diffusion equation for the KOD. The reason the path integral is called a Feynman-Kac formula is that everybody after Kac thinks about going in the opposite direction, starting with the diffusion equation and formulating an equivalent path integral.

Section 4.1 reviews the Cartan-section reduced distribution $C_{T}\left(Zx\right)$, introduces the Harish-Chandra-section reduced distribution $B_{T}\left(Zx\right)$ of Equation (Equation 24), and shows that these two are related by a positive gauge transformation. Section 4.2 defines a normalized version of the Harish-Chandra reduced distribution, denoted by ${\tilde{B}}_{T}\left(Zx\right)$, and finds its path-integral expression in terms of a modified path-integration measure in which the outcome increments are correlated. Section 4.3 formulates the diffusion equations for $B_{T}\left(Zx\right)$ and ${\tilde{B}}_{T}\left(Zx\right)$, and Section 4.4 solves the path integral for ${\tilde{B}}_{T}\left(Zx\right)$ using the stochastic integrals for the Harish-Chandra phase-space coordinates.

### 4.1. Feynman-Kac Formulas

The *ur* KOD of Equation (Equation 235),
(237)$$D_{T}\left(x\right)\equiv \int D\mu \;\left[dw_{\left[0,T\right)}\right]\delta \left(x,\gamma \;\left[dw_{\left[0,T\right)}\right]\right)\;,$$
unravels $Z_{T}$ over the universal domain of G=IWH,
(238)$$Z_{T}=\int_{G}d^{7}\;\mu \left(x\right)\;D_{T}\left(x\right)\;O\;\cdot \;\left(x\right)\;,$$

Reduced distributions are defined on G/Z=RIWH. The first of these reduced distributions, introduced in Equation (Equation 216) as the Cartan-section reduced distribution,
(239)$$C_{T}\left(Zx\right)\equiv \int_{Z}\;d\phi \;d\ell \;D_{T}\left(x\right)e^{-2\ell }\;,$$
can also be defined by the Feynman-Kac path integral,
(240)$$C_{T}\left(Zx\right)=\int D\mu \;\left[dw_{\left[0,T\right)}\right]e^{-2\ell [dw_{\left[0,T\right)}]}\;\delta \left(Zx,Z\gamma \;\left[dw_{\left[0,T\right)}\right]\right)\;,$$
where $\ell \left[dw_{\left[0,T\right)}\right]=\ell_{T}$ is the solution of the SDE of Equation (195) for the Cartan center coordinate *ℓ*,
(241)$$\begin{array}{c} -\ell \left[dw_{\left[0,T\right)}\right]=\int_{0}^{T_{-}}\;\left(\left(\coth2\kappa t-2|\beta_{t}{|}^{2}\;\right)\kappa \;dt+\beta_{t}\sqrt{\kappa }\;dw_{t}^{*}+\beta_{t}^{*}\sqrt{\kappa }\;dw_{t}\right)\;, \end{array}$$
with the notation here emphasizing that this solution is a functional of the sample path of Wiener outcome increments. As was noted in Equation (Equation 198), one can use the transformation from Harish-Chandra coordinates to write
(242)$$\begin{array}{c} -\ell \left[dw_{\left[0,T\right)}\right]=f_{T}-s\left[dw_{\left[0,T\right)}\right]\;, \end{array}$$
where $f_{T}$ is the function of G/Z=RIWH given in Equation (Equation 162),
(243)$$\begin{array}{c} 2f\left(Zx\right)=e^{-r/2}{\sinh\left(r/2\right)\;|\beta +\alpha |}^{2}+e^{-r/2}\cosh\left(r/2\right)\;{\left|\beta -\alpha \right|}^{2}\;, \end{array}$$
with the ruler and the phase-plane coordinates evaluated at time *T*, and
(244)$$\begin{array}{c} -2s_{T}=-2s\left[dw_{\left[0,T\right)}\right]=\int_{0}^{T_{-}}\;\;\int_{0}^{T_{-}}\;\;\kappa \;dw_{t}^{*}\;dw_{s}\;e^{-2\kappa |t-s|} \end{array}$$
is the stochastic integral for the Harish-Chandra center coordinate *s*, given in Equation (Equation 152) and derived in Appendix C. The function $C_{T}\left(Zx\right)$ unravels $Z_{T}$ onto RIWH=G/Z as in Equation (215),
(245)$$\begin{array}{c} Z_{T}=\int_{-1.5ptG/Z}\;\;\;\;d^{5}\;\mu \left(Zx\right)\;C_{T}\left(Zx\right)\;O\;\cdot \;\;\left(D_{\beta }\;e^{-H_{o}r}D_{\alpha }^{†}\right)\;, \end{array}$$
and $C_{T}\left(Zx\right)$ satisfies the FPK equation displayed in Equation (Equation 221),
(246)$$\frac{1}{\kappa }\frac{\partial }{\partial t}C_{t}\left(Zx\right)=\left(-2\partial_{r}-2\coth r+\nabla^{*}\nabla \right)C_{t}\left(Zx\right)\;.$$

There is another natural reduced distribution, conjugate to the Harish-Chandra section,
(247)BT(Zx)≡∫ZdψdsDT(x)e−2s(248)=∫Dμdw[0,T)e−2s[dw[0,T)]δZx,Zγdw[0,T).
This *Harish-Chandra reduced distribution function*, $B_{T}\left(Zx\right)$, unravels $Z_{T}$ as
(249)$$\begin{array}{c} Z_{T}=\int_{-1.5ptG/Z}\;\;\;\;d^{5}\;\mu \left(Zx\right)\;B_{T}\left(Zx\right)\;O\;\cdot \;\;\left(e^{a^{†}\nu }e^{-H_{o}r}e^{a\mu^{*}}\right)\;. \end{array}$$
The two reduced distribution functions are equivalent to one another through a positive gauge transformation [19],
(250)CT(Zx)=∫ZdϕdℓDT(x)e−2ℓ(251)=∫ZdψdsDT(x)e−2[s−f(Zx)](252)=e2f(Zx)BT(Zx).
In the lingo of Feynman-Kac formulas [19], 2f(Zx) is the “convective pressure”.

Section 3.5 introduced the Cartan reduced distribution $C_{T}\left(Zx\right)$ and showed that it is the distribution that expresses POVM completeness. The price for relevance to POVM completeness is that $C_{T}\left(Zx\right)$ has ill-defined normalization and is disconnected from the stochastic-integral solutions for the phase-space variables. The next three sections investigate the Harish-Chandra reduced distribution $B_{T}\left(Zx\right)$. It is clear that $B_{T}\left(Zx\right)$ is not the right distribution for addressing POVM completeness because of the Gaussian gauge function $e^{-2f(Zx)}$, but this Gaussian gauge transformation is just what is needed to get a Gaussian distribution function that can be normalized and whose normalized version can be evaluated from the moments of the phase-space variables, albeit, as we shall see, moments defined relative to a modified path-integration measure.

### 4.2. Normalized Harish-Chandra Reduced Distribution Function and Modified Path-Integration Measure

The Feynman-Kac formula for $B_{T}\left(Zx\right)$, shown in Equation (248), suggests that we combine $e^{-2s[dw_{\left[0,T\right)}]}$ with the Weiner measure of Equation (Equation 56),
(253)$$\begin{array}{cc} D\mu \left[dw_{\left[0,T\right)}\right]e^{-2s[dw_{\left[0,T\right)}]} & =\left(\prod_{k=0}^{T/dt-1}\;d^{\;2}\;\left(dw_{kdt}\right)\right)\;\;{\left(\frac{1}{\pi dt}\right)}^{T/dt}\exp\;\left(-\int_{0}^{T_{-}}\frac{|dw_{t}{|}^{2}}{dt}+\int_{0}^{T_{-}}\;\;\int_{0}^{T_{-}}\;\;\kappa \;dw_{t}^{*}\;dw_{s}\;e^{-2\kappa |t-s|}\right)\;. \end{array}$$
The quadratic functional in the exponential can be written as
(254)−∫0T−|dwt|2dt+∫0T−∫0T−κdwt*dwse−2κ|t−s|=−1dt∫t=0T−∫s=0T−dwt*dwsδts−κdte−2κ|t−s|(255)=−1dt∑k=0N−1∑l=0N−1dwk*dwlδkl−κdte−2κdt|k−l|.
When converting between stochastic integrals and sums, we use $t_{k}=k\;dt$ ($t_{N}=T=N\;dt$) and $dw_{k}=dw_{kdt}=dw_{t_{k}}$.

Now we define the real, symmetric, and positive N×N matrix $M_{T}$, whose matrix elements are
(256)$$\begin{array}{c} M_{kl}=\delta_{kl}-\kappa \;dt\;e^{-2\kappa \;dt\;|k-l|}\;, \end{array}$$
It is elegant for various formal expressions to introduce a continuous version of $M_{T}$, but we do not bother with that here since we work with the sums that the stochastic integrals represent. Notice that $M_{T}$ is a Toeplitz matrix; that is, $M_{k+j,l+j}=M_{kl}$. Putting this together, we have
(257)$$\begin{array}{cc} D\mu \;\left[dw_{\left[0,T\right)}\right]e^{-2s[dw_{\left[0,T\right)}]} & =\left(\prod_{k=0}^{N-1}\;d^{\;2}\;\left(dw_{kdt}\right)\right){\left(\frac{1}{\pi dt}\right)}^{N}\exp\;\left(-\frac{1}{dt}\sum_{k=0}^{N-1}\sum_{l=0}^{N-1}dw_{k}^{*}\;M_{kl}\;dw_{l}\right)\;, \end{array}$$
which integrates over the Wiener outcome paths to
(258)$$\begin{array}{c} \int D\mu \;\left[dw_{\left[0,T\right)}\right]e^{-2s[dw_{\left[0,T\right)}]}={\left(\frac{1}{\pi dt}\right)}^{N}\frac{\pi^{N}}{\det(M_{T}/dt)}=\frac{1}{\det M_{T}}\;. \end{array}$$

This prompts us to define a new (normalized, zero-mean) Gaussian measure on the Wiener outcome paths,
(259)DμMdw[0,T)≡detMTDμdw[0,T)e−2s[dw[0,T)](260)=∏k=0N−1d2dwkdetMT(πdt)Nexp−1dt∑k=0N−1∑l=0N−1dwk*Mkldwl.
Relative to this measure, the outcome increments are correlated,
(261)$$\begin{array}{c} {\langle dw_{k}^{*}\;dw_{l}\rangle }_{M}=dt\;{\left(M_{T}^{\;-\;1}\right)}_{kl}\;. \end{array}$$
Notice that the increment correlation ${\langle dw_{k}^{*}\;dw_{l}\rangle }_{M}$, with *k* and *l* held fixed, changes as the total time *T* changes. The inverse matrix $M_{T}^{\;-\;1}$ matrix is real, symmetric, and positive, all properties inherited from $M_{T}$. The inverse does not inherit the Toeplitz property of $M_{T}$. That $M_{T}$ is Toeplitz implies that it is persymmetric, that is, symmetric across the anti-diagonal. $M_{T}^{\;-\;1}$ does inherit the persymmetry, which turns out to have an important consequence in Section 4.4.

Returning to the Harish-Chandra reduced distribution of Equation (Equation 247), we see that its normalization can be written in several ways: (262)NT≡∫d5μ(Zx)BT(Zx)(263)=∫d5μ(Zx)CT(Zx)e−2f(Zx)(264)=∫Zd7μ(x)DT(x)e−2s(265)=∫Dμdw[0,T)e−2s[dw[0,T)]=1detMT.
The normalized version of the Harish-Chandra reduced distribution is
(266)$$\begin{array}{c} {\tilde{B}}_{T}\left(Zx\right)\equiv \frac{1}{N_{T}}B_{T}\left(Zx\right)=\frac{e^{-2f(Zx)}}{N_{T}}C_{T}\left(Zx\right)\;. \end{array}$$
Section 4.3 finds the diffusion equations satisfied by the reduced Harish-Chandra distributions, and Section 4.4 uses the path integral for the normalized distribution,
(267)(B˜T(Zx)=∫DμMdw[0,T)δZx,Zγdw[0,T),)
to evaluate ${\tilde{B}}_{T}\left(Zx\right)$ from the SDE solutions for the Harish-Chandra coordinates.

### 4.3. Diffusion Equation for Harish-Chandra Reduced Distribution Function

The easiest way to get to the diffusion equation for $B_{t}\left(Zx\right)$ is to return to the FPK equation for $D_{t}\left(x\right)$ given in Equation (Equation 103), write the FPK forward generator Δ in Harish-Chandra coordinates, and then marginalize over the center to get a PDE for $B_{t}\left(Zx\right)$.

By using the frame transformations shown in Equation (Equation 164), it is easy to express the terms of the forward generator in Harish-Chandra coordinates,
(268)2Ho←=−2∂r+2ν1∂ν1+ν2∂ν2(269)=−2∂r−4+2∂ν1ν1+∂ν2ν2,
(270)12Q←Q←=12∇1−12ν1∂s+12ν2∂ψ2(271)=12∇1−12ν1∂s2+12∂ψAQ(272)=12∇12−12∂s∇1ν1+ν1∇1+14ν12∂s2+12∂ψAQ(273)=12∇12−12∂s2∇1ν1−1+14ν12∂s2+12∂ψAQ(274)=14∂s1+12ν12∂s+12∇12−12∂s∇1ν1+12∂ψAQ,
(275)12P←P←=12∇2−12ν2∂s−12ν1∂ψ2(276)=12∇2−12ν2∂s2+12∂ψAP(277)=14∂s1+12ν22∂s+12∇22−12∂s∇2ν2+12∂ψAP,
where the terms
(278)AQ=ν2∇1−12ν1∂s+14ν22∂ψ,(279)AP=−ν1∇2−12ν2∂s−14ν12∂ψ,
are independent of ψ and commute with $\partial_{\psi }$. Putting these expressions together, the FPK forward generator of Equation (Equation 101) becomes in Harish-Chandra coordinates,
(280)(Δ=−2∂r−4+12∂s1+ν12+ν224∂s+12∇12+∇22+2∂ν1ν1+2∂ν2ν2−12∂s∇1ν1+∇2ν2+∂ψA,)
where $A=\frac{1}{2}\left(A_{Q}+A_{P}\right)$.

To derive an evolution equation for $B_{t}\left(Zx\right)$ from Equation (Equation 247), one follows the procedure outlined for $C_{t}\left(Zx\right)$ in Equation (Equation 221), using here the rules that integration by parts on *s* and ψ makes the substitutions $\partial_{s}\to 2$ and $\partial_{\psi }\to 0$. The resulting PDE for $B_{t}\left(Zx\right)$ is
(281)$$\begin{array}{c} \frac{1}{\kappa }\frac{\partial }{\partial t}B_{t}\left(Zx\right)=\left(-2\partial_{r}-3+\frac{\nu_{1}^{2}+\nu_{2}^{2}}{2}+\tilde{\Delta }\right)B_{t}\left(Zx\right)\;, \end{array}$$
where, for brevity, we define a reduced generator for the phase-space-variable derivatives,
(282)Δ˜≡2∂ν1−∇1ν1+2∂ν2−∇2ν2+12∇12+∇22(283)=2∂ν−∇ν+2∂ν*−∇*ν*+∇*∇.
Notice that
(284)2∂νj−∇j=∂νj−e−r∂μj,(285)2∂ν−∇=∂ν−e−r∂μ.

Converting fully to complex phase-space coordinates puts the PDE in the form
(286)(1κ∂∂tBt(Zx)=−2∂r−3+|ν|2+Δ˜Bt(Zx).)
This PDE is ballistic in the ruler *r*—solution proportional to δ(r−2κt)—and Gaussian-preserving in the phase-space variables, but as a consequence of the term $-3+{\left|\nu \right|}^{2}$, it does not preserve normalization [15,16,19]. The effect of the positive gauge transformation from $C_{t}\left(Zx\right)$ to $B_{t}\left(Zx\right)$ is twofold: (i) the norm-nonconserving “potential” term changes character, from a ruler-dependent −2cothr in the PDE for $C_{t}\left(Zx\right)$ in Equation (Equation 246) to a term $-3+{\left|\nu \right|}^{2}$ in the PDE for $B_{t}\left(Zx\right)$ in Equation (Equation 286), which depends on the posterior phase-space variable ν; (ii) there are first-derivative, “vector-potential” terms in the PDE for $B_{t}\left(Zx\right)$, corresponding to the Ornstein-Uhlenbeck behavior of ν in Equation (Equation 148), whereas there are no such terms in the PDE for $C_{T}\left(Zx\right)$.

We turn now to the task of converting the PDE for $B_{t}\left(Zx\right)$ in Equation (Equation 286) to a PDE for the normalized distribution ${\tilde{B}}_{t}\left(Zx\right)$. The initial condition for the PDE in Equation (Equation 286) comes from inserting the Harish-Chandra $D_{dt}\left(x\right)$ of Equation (Equation 472) into $B_{dt}\left(Zx\right)$ as it is expressed in the integral in Equation (Equation 247) specialized to T=dt. Taking the limit dt→0, one finds the expected result that $B_{0}\left(Zx\right)$ is the δ-function on G/Z of Equation (A143):(287)$$\begin{array}{c} B_{0}\left(Zx\right)=\delta \left(Zx,Z1\right)=\delta \left(r\right)\;2\pi \delta^{2}\left(\nu \right)\;2\pi \delta^{2}\left(\mu \right)\;. \end{array}$$
This is expected because the identity is represented uniquely in Harish-Chandra coordinates by ψ=s=r=0, ν=μ=0. The initial condition means that $B_{t}\left(Zx\right)$ is initially normalized to unity; thus the normalization factor $N_{t}$ has the initial value $N_{0}=1$, and the normalized distribution has the same initial condition,
(288)$$\begin{array}{c} {\tilde{B}}_{0}\left(Zx\right)=B_{0}\left(Zx\right)=\delta \left(Zx,Z1\right)\;. \end{array}$$

The normalization factor,
(289)$$\begin{array}{c} N_{T}\equiv \int d^{5}\;\mu \left(Zx\right)\;B_{T}\left(Zx\right)=\int e^{2r}\;dr\;\frac{d^{2}\nu }{2\pi }\;\frac{d^{2}\mu }{2\pi }\;B_{T}\left(Zx\right)\;, \end{array}$$
satisfies the differential equation,
(290)1κdNtdt=∫e2rdrd2ν2πd2μ2π−2∂r−3+|ν|2Bt(Zx)(291)=Nt∫d5μ(Zx)1+|ν|2B˜t(Zx),
where the reader should notice that integration by parts on the ruler becomes the rule $\partial_{r}\to -2$. This differential equation assumes the form
(292)$$\begin{array}{c} \frac{1}{\kappa }\frac{d\ln N_{t}}{dt}=1+n_{t}\;, \end{array}$$
where
(293)$$\begin{array}{c} n_{t}\equiv \langle |\nu_{t}{|}^{2}\rangle_{M}=\int d^{5}\;\mu \left(Zx\right)\;{\left|\nu \right|}^{2}{\tilde{B}}_{t}\left(Zx\right) \end{array}$$
is the second moment of ν relative to the normalized distribution ${\tilde{B}}_{t}\left(Zx\right)$. We place a subscript *M* on this moment because we can use the path-integral expression for ${\tilde{B}}_{T}\left(Zx\right)$ given in Equation (Equation 267) to re-express the moment as
(294)$$\begin{array}{c} n_{T}=\langle |\nu_{T}{|}^{2}\rangle_{M}=\int D\mu_{M}\;\left[dw_{\left[0,T\right)}\right]\;{\left|\nu \left[dw_{\left[0,T\right)}\right]\right|}^{2}\;, \end{array}$$
where $\nu \left[dw_{\left[0,T\right)}\right]=\nu_{T}$ is the stochastic-integral solution for ν, given in Equation (Equation 148). Once the stochastic integral is plugged into this equation, the correlations of the Wiener increments are evaluated according to the modified measure $D\mu_{M}\;\left[dw_{\left[0,T\right)}\right]$, that is, as in Equation (Equation 261).

The PDE for the normalized reduced distribution ${\tilde{B}}_{t}\left(Zx\right)$ of Equation (Equation 266) now follows as
(295)$$\begin{array}{c} \frac{1}{\kappa }\frac{\partial }{\partial t}{\tilde{B}}_{t}\left(Zx\right)=\left(\;-\frac{1}{\kappa }\frac{d\ln N_{t}}{dt}-2\partial_{r}-3+{\left|\nu \right|}^{2}+\tilde{\Delta }\right){\tilde{B}}_{t}\left(Zx\right)\;. \end{array}$$
Inserting Equation (Equation 292) gives
(296)(1κ∂∂tB˜t(Zx)=−2∂r−4+|ν|2−nt+Δ˜B˜t(Zx).)
It is easy to see how this equation preserves normalization. The presence of the moment $n_{t}$, essential for normalization, makes the equation nonlinear, but it can still be solved easily.

To solve for ${\tilde{B}}_{t}\left(Zx\right)$, one notes that the PDE is ballistic in the ruler *r* and Gaussian-preserving in the phase-space variables. Thus, the solution has the form ${\tilde{B}}_{t}\left(Zx\right)=e^{-2r}\delta \left(r-2\kappa t\right)\Phi_{t}\left(Zx\right)$, where $\Phi_{t}\left(Zx\right)$ is a normalized, zero-mean Gaussian in the phase-space variables ν and μ. The derivatives in the PDE of Equation (Equation 296) are invariant under complex conjugation and under simultaneous rephasing of the phase-space variables, that is, $\nu \to \nu e^{i\chi }$ and $\mu \to \mu^{i\chi }$. It is productive to think of the invariance under complex conjugation as invariance under the change of phase-space coordinates $\nu ↔\nu^{*}$ and $\mu ↔\mu^{*}$. It is useful to appreciate that all the diffusion equations in this paper share these invariance properties. The invariance under simultaneous rephasing implies that $\Phi_{t}\left(Zx\right)$ is a zero-mean Gaussian since it starts from a zero-mean δ-function initial condition. It further implies that $\Phi_{t}\left(Zx\right)$ is determined by the three nonzero second moments of the phase-space variables: $n_{t}$ of Equation (Equation 293) and
(297)mt≡〈|μt|2〉M=∫d5μ(Zx)|μ|2B˜t(Zx),(298)qt≡〈νt*μt〉M=〈μt*νt〉M=∫d5μ(Zx)ν*μB˜t(Zx),
with the invariance under complex conjugation implying that $q_{t}$ is real. This form of the solution for ${\tilde{B}}_{t}\left(Zx\right)$ in hand, one derives from the PDE of Equation (Equation 296) first-order temporal differential equations for the second moments $n_{t}$, $m_{t}$, and $q_{t}$. Just as the ordinary differential equation (ODE) for the normalization factor involves a second moment, so the equations for the second moments involve fourth moments. The Gaussian form of the solution relates the fourth moments to second moments, thus closing the system of differential equations. The resulting three ODEs, for the derivatives of $n_{t}$, $m_{t}$, and $q_{t}$, have terms that are constant, linear, and quadratic in the moments; the presence of the quadratic terms makes these ODEs (coupled) Riccati equations. The last step is to solve the three Riccati equations with initial conditions $n_{0}=m_{0}=q_{0}=0$, which are implied by the δ-function initial condition for ${\tilde{B}}_{0}\left(Zx\right)$. This gives the solution for ${\tilde{B}}_{t}\left(Zx\right)$. By integrating to find $N_{T}$, using the solution for $n_{t}$, one can backtrack to find $B_{T}\left(Zx\right)$ and $C_{T}\left(Zx\right)$ from Equation (Equation 266).

We have carried out this procedure of deriving the Riccati equations from the PDE in Equation (Equation 296), but we do not present that derivation in this paper, preferring instead to use a different method, which derives the Riccati equations from the path integral for ${\tilde{B}}_{T}\left(Zx\right)$. Implementing that method is the final task of this paper, carried out in the next section.

### 4.4. Normalized Harish-Chandra Reduced Distribution Function from Its Path Integral

The path integral for ${\tilde{B}}_{T}\left(Zx\right)$, displayed in Equation (Equation 267), can be written explicitly in terms of the δ-function in Harish-Chandra coordinates,
(299)B˜T(Zx)=∫DμMdw[0,T)e−2rδ(r−rT)2πδ2(ν−νT)2πδ2(μ−μT)(300)=e−4κTδ(r−2κT)∫DμMdw[0,T)2πδ2ν−ν[dw[0,T)]2πδ2μ−μ[dw[0,T)],
where $r_{T}=2\kappa T$, $\nu_{T}=\nu \left[dw_{\left[0,T\right)}\right]$, and $\mu_{T}=\mu \left[dw_{\left[0,T\right)}\right]$ are the solutions to the SDEs for the ruler and the Harish-Chandra phase-space coordinates, shown in Equations (Equation 147)–(Equation 149). The measure for the path integral is a (normalized, zero-mean) Gaussian measure in the outcome increments $dw_{t}$; thus, the path integral gives a normalized, zero-mean Gaussian in ν and μ, which is determined by the three (real) moments introduced in the previous section:(301)nT=〈|νT|2〉M=∫d5μ(Zx)|ν|2B˜T(Zx)=∫DμMdw[0,T)|ν[dw[0,T)]|2,(302)mT=〈|μT|2〉M=∫d5μ(Zx)|μ|2B˜T(Zx)=∫DμMdw[0,T)|μ[dw[0,T)]|2,(303)qT=〈νT*μT〉M=〈μT*νT〉=∫d5μ(Zx)ν*μB˜T(Zx)=∫DμMdw[0,T)ν[dw[0,T)]*μ[dw[0,T)].
In this context, that the first moments and all the other second moments of the phase-space variables are zero follows from the fact that the measure is invariant under simultaneous rephasing of all the outcome increments; the reality of $q_{T}$ follows from the fact that the measure is unchanged under the transformation $dw_{\left[0,T\right)}\to dw_{\left[0,T\right)}^{*}$. These properties come from the fact that $M_{T}$ is real and symmetric.

Plugging in the stochastic-integral solutions for $\nu_{T}$ and $\mu_{T}$ puts these moments into the following form: (304)nT=〈|νT|2〉M=κ∑k,l=0N−1〈dwk*dwl〉Me−2κ(T−tk)e−2κ(T−tl)(305)=κdt∑k,l=0N−1e−2κdt(N−k)(MT−1)kle−2κdt(N−l),(306)mT=〈|μT|2〉M=κ∑k,l=0N−1〈dwk*dwl〉Me−2κtke−2κtl(307)=κdt∑k,l=0N−1e−2κdtk(MT−1)kle−2κdtl,(308)qT=〈νT*μT〉M=〈μT*νT〉M=κ∑k,l=0N−1〈dwk*dwl〉Me−2κ(T−tk)e−2κtl(309)=κdt∑k,l=0N−1e−2κdt(N−k)(MT−1)kle−2κdtl.
In the final form of $q_{T}$, it is evident that $q_{T}$ is real. Notice that these expressions satisfy the zero initial conditions.

That $M_{T}$ is a Toeplitz matrix introduces an additional, quite important symmetry. The inverse matrix does not inherit the Toeplitz property of *M*, but it does inherit a less restrictive property. That $M_{T}$ is Toeplitz implies that it is symmetric about the anti-diagonal; that is, $M_{kl}=M_{N-1-l,N-1-k}$. A matrix that is symmetric about the anti-diagonal is called *persymmetric*. It is easy to show that the inverse of a persymmetric matrix is persymmetric, so $M_{T}^{\;-\;1}$ satisfies ${\left(M_{T}^{\;-\;1}\right)}_{kl}={\left(M_{T}^{\;-\;1}\right)}_{N-1-l,N-1-k}$. Persymmetry has a major consequence for the three moments, which comes from manipulating $m_{T}$:(310)mT=κdt∑k,l=0N−1e−2κdt(N−1−k)(MT−1)N−1−k,N−1−le−2κdt(N−1−l)(311)=e4κdtκdt∑k,l=0N−1e−2κdt(N−k)(MT−1)lke−2κdt(N−l).
We can set $e^{4\kappa \;dt}=1$ and thus conclude that
(312)$$\begin{array}{c} m_{T}=n_{T}\;. \end{array}$$
The complementarity in time of $\nu_{T}$ and $\mu_{T}$ was discussed in Section 3.2: the post-measurement variable $\nu_{T}$ of Equation (Equation 148) depends exponentially on the end of the outcome register, and the POVM variable $\mu_{T}$ of Equation (Equation 149) depends exponentially on the beginning of the register. The persymmetry of $M_{T}$ and $M_{T}^{\;-\;1}$ expresses that the beginning and end of the record look the same statistically, so it is not surprising that the persymmetry implies that $m_{T}=n_{T}$.

The next step, deriving Ricatti equations for the three moments, involves incrementing the moments from *T* to T+dT. The tedious part of this task is determining how $M_{T}^{\;-\;1}$ increments—that is, finding $M_{T+dT}^{\;-\;1}$—and that can be done using the Schur complement. We relegate this entire task to Appendix G and here skip directly to the coupled Ricatti ODEs, taken from Equations (A174)–(A176): (313)1κdnTdT=(1−nT)2,(314)1κdmTdT=qT+e−2κT2,(315)1κdqTdT=−qT(1−nT)+e−2κT(1+nT).
With the zero initial conditions, these have the solutions,
(316)nT=mT=κT1+κT,(317)qT=11+κT−e−2κT=−κT1+κT+2e−κTsinhκT.

It is quite instructive to notice that the equality $n_{T}=m_{T}$ and the reality of $q_{T}$ together imply that the sum and difference Harish-Chandra phase-space variables are uncorrelated,
(318)$$\begin{array}{c} \langle {\left(\nu_{T}\pm \mu_{T}\right)}^{*}{(\nu_{T}\mp \mu_{T}\rangle }_{M}=\langle |\nu_{T}{|}^{2}\rangle_{M}-\langle |\mu_{T}{{|}^{2}\rangle }_{M}\mp {\langle \nu_{T}^{*}\mu_{T}\rangle }_{M}\pm {\langle \mu_{T}^{*}\nu_{T}\rangle }_{M}=0\;, \end{array}$$
with second moments,
(319)$$\begin{array}{c} \langle |\nu_{T}\pm \mu_{T}{|}^{2}\rangle_{M}=\langle |\nu_{T}{|}^{2}\rangle_{M}+\langle |\mu_{T}{{|}^{2}\rangle }_{M}\pm {\langle \nu_{T}^{*}\mu_{T}\rangle }_{M}\pm {\langle \mu_{T}^{*}\nu_{T}\rangle }_{M}=2\left(n_{T}\pm q_{T}\right)\;, \end{array}$$
where
(320)nT+qT=2e−κTsinhκT,(321)nT−qT=−1+κT1+κT+e−2κT=2e−κTcoshκTΣT1+κT,

The width $\Sigma_{T}=\kappa T-\tanh\kappa T$ was introduced in Equation (Equation 228). It is worth noting the early- and late-time behavior of the various moments:(322)nT=mT=κT,κT≪1,1,κT≫1,(323)qT=κT,κT≪1,1/κT,κT≫1,(324)nT+qT=2κT,κT≪1,1,κT≫1,(325)nT−qT=23(κT)3,κT≪1,1,κT≫1.
At early times, ν and μ are tightly correlated, with their sum undergoing standard diffusion; at late times, they become uncorrelated, and their second moments saturate at 1.

The sum and difference variables being uncorrelated, the Gaussian path integral from Equation (Equation 299) has the normalized solution,
(326)$$\begin{array}{cc} {\tilde{B}}_{T}\left(Zx\right) & =e^{-4\kappa T}\delta \left(r-2\kappa T\right)\;4\pi \frac{1}{2\pi (n_{T}+q_{T})}\exp\;\left(-\frac{{\left|\nu +\mu \right|}^{2}}{2(n_{T}+q_{T})}\right)\;4\pi \frac{1}{2\pi (n_{T}-q_{T})}\exp\;\left(-\frac{{\left|\nu -\mu \right|}^{2}}{2(n_{T}-q_{T})}\right)\;. \end{array}$$
Noting that
(327)$$\begin{array}{c} \left(n_{T}+q_{T}\right)\left(n_{T}-q_{T}\right)=\sinh2\kappa T\;\frac{2\Sigma_{T}}{e^{2\kappa T}\left(1+\kappa T\right)}\;, \end{array}$$
we can put the normalized solution in the form
(328)$$\begin{array}{cc} {\tilde{B}}_{T}\left(Zx\right) & =\frac{1}{\sinh2\kappa T}\;\delta \left(r-2\kappa T\right)\;\frac{2e^{-2\kappa T}\left(1+\kappa T\right)}{\Sigma_{T}}\exp\;\left({-|\nu +\mu |}^{2}\frac{e^{\kappa T}}{4\sinh\kappa T}-{\left|\nu -\mu \right|}^{2}\frac{e^{\kappa T}}{4\cosh\kappa T}\frac{1+\kappa T}{\Sigma_{T}}\right)\;, \end{array}$$
which satisfies the δ-function initial condition of Equation (Equation 288).

To retrieve the unnormalized distribution $B_{T}\left(Zx\right)=N_{T}{\tilde{B}}_{T}\left(Zx\right)$, one needs $\det M_{T}=1/N_{T}$, which, according to Equation (Equation 501) or Equation (Equation 292), satisfies the equation
(329)$$\begin{array}{c} \frac{1}{\kappa }\frac{d\ln\det M_{T}}{dT}=-\left(1+n_{T}\right)=-2+\frac{1}{1+\kappa T}\;, \end{array}$$
with the solution, for the initial condition $\det M_{0}=1$,
(330)$$\begin{array}{c} \det M_{T}=e^{-2\kappa T}\left(1+\kappa T\right)=\frac{1}{N_{T}}\;. \end{array}$$
The unnormalized distribution is therefore
(331)BT(Zx)=1sinh2κTδ(r−2κT)2ΣTexp−|ν+μ|2eκT4sinhκT−|ν−μ|2eκT4coshκT1+κTΣT(332)=1sinh2κTδ(r−2κT)2ΣTexp−|β+α|2e−κTsinhκT−|β−α|2e−κTcoshκT1+κTΣT.
The second line transforms to Cartan phase-space coordinates. Unnormalizing changes the Gaussian’s prefactor; transforming to Cartan coordinates changes the Gaussian. The final step is to undo the gauge transformation to get back to the Cartan reduced distribution,
(333)$$\begin{array}{c} C_{T}\left(Zx\right)=e^{f(Zx)}B_{T}\left(Zx\right)=\frac{1}{\sinh2\kappa T}\;\delta \left(r-2\kappa T\right)\;\frac{2}{\Sigma_{T}}\exp\;\left(-\frac{{\left|\beta -\alpha \right|}^{2}}{\Sigma_{T}}\right)\;, \end{array}$$
which matches the solution in Equation (Equation 227), which was obtained from the PDE for $C_{t}\left(Zx\right)$ displayed in Equation (Equation 221).

It is fair to ask whether the point of this section is just to provide a different, more complicated route to the solution for $C_{T}\left(Zx\right)$. We think it is more than that, and here is why. It all comes back to the elephant in the room, that is, how one handles the normalization or scaling of the Kraus operators that comes from the center *Z*. The Cartan reduced distribution, defined in Equation (Equation 216) and determined from the diffusion equation in Equation (Equation 221), succeeds in representing POVM completeness by marginalizing the *ur*-distribution $D_{T}\left(x\right)$ over the Cartan-center normalization, $e^{-2l}=e^{2f(Zx)}e^{-2s}$; this gives a distribution uniform in the Cartan sum variable β+α and thus spread over all of phase space in a way that gives POVM completeness. As a consequence, however, the moments of $C_{T}\left(Zx\right)$ are not those of the stochastic integrals for the phase-plane variables. The Harish-Chandra reduced distribution $B_{T}\left(Zx\right)$, defined in Equation (Equation 247), marginalizes $D_{T}\left(x\right)$ over the Harish-Chandra-center normalization, $e^{-2s}$. After normalization to unity by the factor $N_{T}$, the normalized distribution ${\tilde{B}}_{T}\left(Zx\right)$ is determined from the diffusion equation shown in Equation (Equation 296) or by applying the path-integral expression of Equation (Equation 267), with its modified path measure, to the stochastic integrals for the Harish-Chandra phase-space variables. The route from ${\tilde{B}}_{T}\left(Zx\right)$ to POVM completeness runs backwards through the normalization factor $N_{T}$ and the anti-Gaussian gauge transformation $e^{2f(Zx)}$ and arrives at $C_{T}\left(Zx\right)$. The point of this section, one might say, is to find and reveal these connections among path integrals, diffusion equations, and SDEs; discovering these connections, driven in this paper by a combination of necessity and opportunity, allows us to re-unite the three faces of the stochastic trinity. Accessing the entire stochastic trinity by using a positive gauge transformation that is grounded in a problem’s Lie group—this, we hope, might be generally applicable to Feynman-Kac formulas for non-normalization-preserving diffusion equations.

## 5. Concluding Remarks. The Stochastic Trinity

We set out on the project of analyzing simultaneous measurements of noncommuting observables [39,40] with the goal of showing that such measurements end up with a POVM in the overcomplete coherent-state basis. We now think that we have uncovered something more ambitious, a distinctive new window into the space of quantum dynamics. The formulation of the problem of continual, differential measurements invites one—compels one, really—to think in terms of the paths of Wiener outcome increments. These outcome paths, as sample paths drawn from the Wiener measure, know nothing about the space in which they are wandering. When they are instantiated in the exponents of Kraus operators, however, the time-ordered products of the differential Kraus operators generate a (complex) Lie-group manifold, the instrumental Lie group, in which the Kraus operators are, in the way of groups, both the transformations and the moving points. The motion in the instrumental Lie-group manifold is described by Kraus-operator SDEs or by the diffusion of the KOD, as embodied in an FPK diffusion equation. The continuous, but not differentiable, paths are handled effortlessly by the Itô calculus of the outcome increments, with its terms of order $\sqrt{dt}$ and dt. This requires getting just beyond the linear structure of vector fields (right-invariant derivatives) and one-forms (right-invariant one-forms) on the Lie-group manifold, as is evident from our discussion of Stratonovich vs. Itô SDEs and in the derivation of the FPK diffusion equation, where right-invariant derivatives end up as diffusive second derivatives. We end up in a very comfortable place, working in all three corners of the stochastic trinity: path integrals, FPK diffusion equations, and SDEs, all three describing motion, equivalently, on the instrumental Lie-group manifold.

Why do others not find the same comfort in all three faces of the trinity? Field theorists, interested in the propagator of closed-system dynamics, have a different way of handling the continuous, but not differentiable paths, coming from a Stratonovich calculus way of dealing with the temporal derivatives in the kinetic terms in a Lagrangian. While they usually have an equivalent Schrödinger-like equation for the propagator, they do not have the analog of SDEs, even though the problem often undergoes a Wick rotation into “Euclidean spacetime”.

Open-systems theorists, both in condensed matter physics and in quantum optics, generally work with master equations or stochastic master equations for the evolution of quantum states and sometimes with diffusion equations for a probability distribution associated with the states. Those who start with diffusion equations can avail themselves of a Feynman-Kac formula for a path integral, but the connection to SDEs has generally not been made. The reason for not using all three faces of the stochastic trinity is, we think, a failure to identify the appropriate Lie-group manifold; this failure is connected to the emphasis on the evolution of quantum states, which obscures nearly entirely the Lie-group manifold on which the open-system dynamics occur.

Suffice it to say that we think we have found something: the home of quantum dynamics, the Lie-group manifold that supports all three faces of the trinity. The exhausted reader who has survived to read this concluding sentence of a very long paper might be pleased—or so we hope—to learn that our ambition is larger than was evident at the beginning.

## Data Availability

This project generated analysis and theory, not data.

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
