# Peer review of "Simultaneous Momentum and Position Measurement and the Instrumental Weyl-Heisenberg Group"

_entropy, 2023, doi:10.3390/e25081221_

Round 1

Reviewer 1 Report

This ambitious paper takes the authors' approach to continuous weak measurements developed in a companion paper (which, by the way, is necessary reading to understand this paper), as a fundamentally new way of formulating non relativistic quantum theory in terms of stochastic processes of informationally complete  measurements. It is difficult to read (the occasional amusing side comments ease the journey) but it amply rewards the effort. Certainly some of the things I thought I knew took on a surprising new  significance as presented here. It was especially satisfying to see the presentation done using the path integral method. This holds out some hope that a relativistic formulation might be possible and this paper might inspire some brave soul to give it a go. That however will require generalising the current treatment to open fermonic systems such as occur in mesoscopic electronics. 

I would be interested to learn what the author's approach would offer to quantum parameter estimation in  continuous measurement ( as Gammelmark and Moller pioneered). This is now assuming considerable significant in the study of fluctuation theorems for quantum stochastic thermodynamics. 

The paper should be accepted in its current form. 

Author Response

Author's response to report of first referee (Manuscript ID: entropy-2457552)
Christopher S. Jackson and Carlton M. Caves

No response is really required, except perhaps to thank this reviewer for a very favorable report.  

We do have big plans for extending our formalism to other approaches, now that the the quite technical foundation has been laid. 

Reviewer 2 Report

In the introduction of the paper is announced that the problem of the simultaneous measurement of position and momentum of quantum system will be studied. In the long text (53 pages), the big number of mathematical formulas and citations of known literatures is presented. Due to this the promised study is difficult to extract from the results of the paperAlso, the authors write: " Before proceeding, we caution that this paper uses a mathematical apparatus not familiar to most physicists and quantum scientists. This apparatus is introduced here naturally, as it becomes both desirable and necessaryReaders who are made uncomfortable by this apparatus are urged to consult the companion paper [40], which attempts to persuade the reader that the unfamiliar mathematical concepts and techniques are essential tools a new way of thinking and doing thinking and doing and then introduces these tools as gently as possible. " Short review of  the mathematical formalism result introduced in [40] must be presented in the text and the correct citation (number, year) of arxive preprint [40] must be given as well as citation of future publication in a journal of this preprint.  I can recommend to publish this paper only if a very simple example of quantum system will be used to demonstrate the main statement of the paper and possibility of simaltaneous measurement of position and momentum wil be clarified for the community. Also in the very end of the paper in Acknowledgement there is a phrase: "Any subjective views or opinions that might be expressed in the paper do not necessarily represent the views of the U.S. Department of Energy or the United States Government." After such improvement on the example of simple quantum model I hope that the U.S. Department of Energy and the United States Government will agree with the result of the paper .   The paper can be written in more short form. Several formulas are repeated and the repetition must  be removed.  

Author Response

Author's response to report of second referee (Manuscript ID: entropy-2457552)
Christopher S. Jackson and Carlton M. Caves

The referee's main complaint is that the companion paper (Ref. 40) was not available at the time of submission of this paper to Entropy.  We do regret this and acknowledge that the order in which we submitted the two papers to Entropy was a mistake.  But the companion paper is now on the arXiv, and the arXiv reference is now in Ref. 40.  The companion paper has been submitted to Entropy and has received two quite favorable reports in the first round of review. 

The referee insists that our paper should be published only if ``a very simple example of quantum system will be used to demonstrate the main statement of the paper and possibility of simultaneous measurement of position and momentum will be clarified for the community.''  The problem with this insistence is that we *are* analyzing the obvious very simple model of simultaneous measurement of position and momentum.  The full analysis of this simple model requires quite sophisticated mathematical techniques.  The failure to apply these techniques to this model in the past is the reason that this foundational problem has not been done successfully before.

The support information that the referee objects to is boilerplate that is required when acknowledging support from Sandia National Laboratories.

The referee objects to the length of the paper and says that there are many redundancies that could be eliminated.  We plead guilty to the redundancies; they were deliberately engineered into this very long paper to make it easier to read.  Eliminating them would not shorten the paper substantially, but it would make the paper less accessible.